



# Analyses of sea surface Chlorophyll-a trends and variability in a period of rapid Climate change, German Bight, North Sea.

Felipe de Luca Lopes de Amorim[1], Areti Balkoni[1], Vera Sidorenko[1], Karen Helen Wiltshire[1]

[1] Alfred-Wegener-Institut Helmholtz-Zentrum für Polar- und Meeresforschung, List auf Sylt, 25992, Germany

*Correspondence to*: Felipe de Luca Lopes de Amorim (felipelamorim@gmail.com)

**Abstract.**

Satellite remote sensing of ocean colour properties allows observation of the ocean with high temporal and spatial coverage, facilitating the better assessment of changes in marine primary production. Ocean productivity is often assessed using satellite derived chlorophyll-a concentrations, a commonly used proxy for phytoplankton concentration. We used the Copernicus GlobColour remote sensing Chl-a surface concentration, comparing with the Helgoland Roads Chl-a in situ data, to investigate seasonal and non-seasonal variability, temporal trends, changes in spring bloom Chl-a magnitude and the relationship with sea

surface temperature (SST) and mixed layer depth (MLD) in the German Bight (GB) from 1998 to 2020. Empirical Orthogonal Functions and Multi Covariance Analysis were employed, in order to investigate dominant spatial and temporal patterns (modes) related to the main processes of Chl-a variability and to extract the dominant structures that maximize the covariance between Chl-a and SST|MLD fields. High levels of Chl-a were found near the coast, showing a decreasing gradient towards offshore waters. A significant long-term positive trend was observed close to the Elbe estuary and adjacent area, while 30%

of the GB was characterized by a significant negative trend. No significant trends were observed during spring blooms, but the distribution of Chl-a anomalies changed when periods from 1998 to 2009 and 2010 to 2020 were compared, mostly showing a shift towards negative anomalies and decrease in variability. The Chl-a non-seasonal variability showed that the first four modes explained around 45% with the first and second modes related to inter and intra-annual variability, respectively, observed in the temporal principal components spectral analyses. Linear trends between Chl-a anomalies and SST and MLD

anomalies were weak and described by opposite signal in offshore and coastal areas, with negative correlation between Chl-a and temperature in offshore areas and positive in mostly coastal areas, while for MLD was the contrary. The monthly chlorophyll-a concentration anomalies covaried 45% with sea surface temperature anomalies and 23% with mixed layer depth anomalies. This study demonstrated that the Copernicus Global Ocean Colour chlorophyll-a concentration product can assess mostly of the known processes connected to chlorophyll-a surface variability in the German Bight The monthly averages were

suitable to investigate long-term trends and variability for SST, but for MLD, higher frequency should be used. The causes for the significant negative trends in most of the central German Bight cannot be solely explained by the direct effect of warming.





However, the rising water temperature, combined with its indirect effects on other variables, can partially explain these observed trends.

## 1 Introduction

Long-term ocean productivity serves as a crucial indicator of planetary change, with direct ties to shifts in ecosystem functionality and the decline of higher trophic levels (Henson et al, 2010; Stock et al, 2014). Marine ecosystems, particularly those in shelf seas, are subject to both natural variability and the increasing stress from anthropogenic climate change. The German Bight, a highly dynamic region of the North Sea, has undergone significant changes over the past sixty years.

Pronounced shifts in seawater nutrient concentrations and stoichiometry are well reported (Raabe and Wiltshire 2009; van Beusekom et al. 2019; Balkoni et al. 2023). In parallel, Sea Surface Temperature (SST) has been on a steady rise since 1962 (Amorim & Wiltshire et al., 2023). Changes in nutrient concentrations have profound impacts on phytoplankton productivity and species composition (Hickel et al., 1993; Topcu et al., 2011; Burson et al, 2016). However, the role of increasing temperature remains unclear.


Increasing SST can affect phytoplankton biomass both directly, by influencing species physiology and ecosystem structure, and indirectly, by altering the hydrographic conditions of a region. For example, increasing SST can enhance phytoplankton cell division rate (Hunter-Cevera et al. 2016). However, if the optimum temperature is exceeded, the growth rate and primary production may decrease (Baker et al., 2016). Indirect effects include changes in regional hydrographic conditions, as rising

temperature can increase water column stratification and reduce the Mixed Layer Depth (MLD), thereby affecting light and nutrient availability to primary producers. Understanding the drivers of phytoplankton biomass variability in coastal waters is crucial for gaining insights into the dynamics and fluctuations of higher trophic level populations (Marrari et al, 2017), and for assessing the ecological status of the coastal environment (European Environment Agency, 2022).

Chlorophyll-a (Chl-a) is commonly used to estimate the phytoplankton biomass in the water column (Eisner et al., 2016; Huot et al., 2007). However, acquiring accurate spatial and temporal data on Chl-a concentration can be a challenge due to data scarcity in one or both domains. While extensive time series data can be collected at a single geographical point, this approach lacks spatial resolution. Satellite data offers a solution to this problem by providing comprehensive spatial and temporal coverage, enabling the assessment of Chl-a spatiotemporal variability. The detection of phytoplankton via remote sensing relies

on the unique properties of chlorophyll, which absorbs and reflects sunlight in the visible-near infrared part of the electromagnetic spectrum.



Surface Chl-a remote sensing products have long been an effective observational methodology in both coastal and open ocean environments (Henson et al., 2009; Henson et al., 2010; Fernández-Tejedor et al., 2022). In open ocean waters, remote sensing

measurements allow for accurate determination of ocean colour. However, in coastal waters, the detection of ocean colour is complicated by the presence of suspended particulate and dissolved matter, making the retrieval of Chl-a concentration more complex in these systems (Pahlevan et al., 2020). One limitation of satellite-derived Chl-a is that it restricts the accurate depiction of the entire system due to the absence of a vertical dimension (Zhao et al., 2019). Despite this, it enables a spatially dynamic description of surface chlorophyll. For shallow seas like the German Bight, remote sensing provides a good

representation of the chlorophyll in the water column, as the first optical depth, ranging from 1 to 12 m in the region of interest, is sampled (Doerffer and Fischer, 1994). Turbulent mixing induced by storms, tides, and internal waves redistributes chlorophyll to near-surface depths (Zhang et al., 2019; Becherer et al., 2022). Overall, remote sensing, despite the limitations, remains a valuable tool for studying phytoplankton biomass in both space and time (Blondeau-Patissier et al., 2014).

In this study, we investigated the long-term trends and variability of Chlorophyll-a (Chl-a) in the German Bight, considering the rapid increase in SST in the region over the past two decades. We retrieved the Copernicus GlobColour-merged Chl-a product, spanning from January 1998 to December 2020, and compared it with in-situ Chl-a measurements. We also examined the spatial and temporal covariability of Chl-a in relation to SST and MLD. Our specific objectives were to understand the following:

(i)       The differences between in-situ and satellite-derived Chl-a.

    (ii)      The seasonal variability of Chl-a.

    (iii)     The long-term trends of Chl-a.

    (iv)     The dominant modes of Chl-a variability.

    (v)      The relationship between Chl-a, SST, and MLD in the region.

By addressing these points, we aim to provide an updated perspective on Chl-a trends and variability in the German Bight, as well as a comprehensive understanding about its relationship with SST and MLD in the study area.



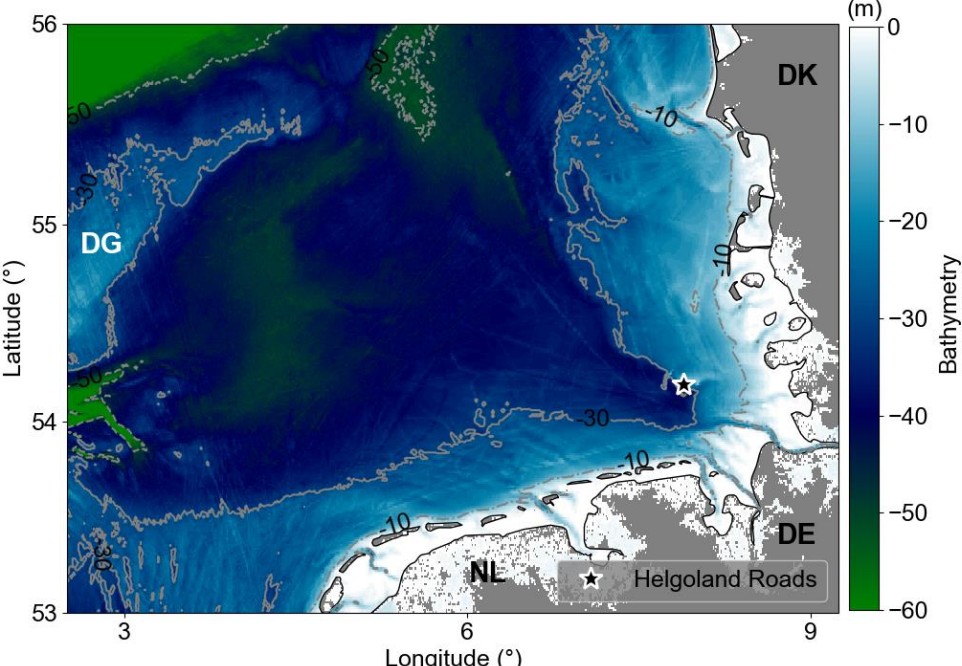

**Figure 1: Bathymetry of the German Bight. Dot-dashed, solid, and dashed grey lines are 10, 30, and 50 m isobaths respectively. Areas with depths less than 5 m represented by white colour. Black star marks the location that the Helgoland Roads time series have been collected. The acronym DG refer to Dogger Bank; DE to Germany; DK in Denmark; and NL to Netherlands.**

## 2 Data and Methods

### 2.1 Study Area

The German Bight is a coastal area in the south-eastern North Sea bounded by the Netherlands (NL), Germany (DE), and Denmark (DK) (Fig.1). The area defined in this work ranges from 2.5-9.25°N and 53-56°E. It extends from the Elbe estuary to the northwest and passes the Dogger Bank (DB) and it has a maximum depth of about 50 m (Fig.1). A 30 to 40 m deep funnel-like feature, defined by the deep Elbe valley, crosses the area in a diagonal direction from southeast to northwest (Stanev et al., 2014). The hydrodynamics of the region are very complex due to the interactions of riverine discharges, central North Sea water, Atlantic water (which is inserted into the region through the English Channel), and the tidal and atmospheric forcing (Becker et al., 1992; Kerimoglu et al., 2020).

The productivity in the German Bight is linked to its hydrographic conditions and bathymetric features (Emeis et al. 2015; Capuzzo et al., 2018). In regions where the depth is less than 20 m, the vertical concentration of Chl-a is largely homogenized



throughout the year, primarily due to the mixing effects of tidal currents and wind. However, in areas where the depth exceeds 20 m, the vertical Chl-a concentration exhibits significant variability in terms of its extent, duration, and intensity (Schrum, 1997; Zhao et al, 2019). This variability is largely attributed to stratification and biogeochemical processes, such as the slow decomposition rates of organic matter (van Beusekom et al., 1999), which result in a distinct vertical Chl-a distribution when compared to shallower regions (Zhao et al, 2019).


## 2.2 Chlorophyll-a Concentration Remote Sensing Data

In this study, we downloaded the Copernicus Marine Service (CMS) GlobColour. daily interpolated cloud-free surface Chl-a concentration product, ranging from January 1998 to December 2020. This product was available for download at the time of access on 20 October 2021 from https://resources.marine.copernicus.eu/ under the product described as

OCEANCOLOUR_ATL_CHL_L4_REP_OBSERVATIONS_009_098. This Chl-a dataset was produced using multiple sensors (multi-sensor product), multiple Chl-a algorithms and a daily space-time interpolation scheme, with a 1 km$^2$ spatial resolution (Garnesson et al, 2019).

## 2.3 Sea surface temperature, mixed layer depth and North Atlantic Oscillation (NAO)

Daily fields of sea surface temperature (SST) and mixed layer depth (MLD) were obtained from CMS (https://data.marine.copernicus.eu/), spanning from January 1998 to December 2020. The SST dataset is gap-free maps of daily average SST at 0.05deg. horizontal grid resolution, using satellite data from the Advanced Along-Track Scanning Radiometer (AATSR), Sea and Land Surface Temperature Radiometer (SLSTR) and the Advanced Very High Resolution Radiometer (AVHRR) (Lavergne et al., 2019; Merchant et al., 2019; Good et al., 2020).

Daily mixed-layer depth data (≈7 km horizontal resolution) were obtained from CMS (https://data.marine.copernicus.eu/), as part of the Atlantic-European North West Shelf-Ocean Physics Reanalysis product. The MLD was defined as the depth where the density increase compared to density at 3m depth corresponds to a temperature decrease of 0.2°C in local surface conditions (Kara, 2000; PUM, 2021).

The NAO winter index data was obtained from the Climate Analysis Section, National Center for Atmospheric Research

(NCAR). It is calculated as the leading Empirical Orthogonal Function of sea level pressure anomalies considering the gradient between the Icelandic low and Azores high (Hurrell et al., 2003). The winter NAO index is the mean of the index for the December, January and February months.

## 2.4 In situ data

In-situ Chl-a concentrations, measured with a high-performance liquid chromatography (HPLC) on a work-daily basis since

2004 at Helgoland Roads (see black star in Fig. 1 for the data site) (Wiltshire et al., 2008), were used for the evaluation of the





satellite derived product. Helgoland Island is located in the German Bight approximately 60 km of the German coast and since 1962, surface water samples have been collected at the Helgoland Roads site, between the Helgoland and Düne Islands (54° 11.3′ N, 7°54.0′ E). The samples are representative for the whole water column due to the well-mixed conditions (Wiltshire et al., 2010). The Helgoland Island is in a transition zone in the German Bight, influenced by offshore (higher salinity) and coastal

waters (lower salinity) (Wiltshire et al., 2015).

## 2.5 Data pre-processing

As we were interested in long-term trends and variability, Chl-a, SST and MLD daily fields were monthly averaged, also avoiding problems with spatial missing data. We computed monthly anomalies by subtracting the monthly means absolute

values by the climatological averages, which are the mean of monthly absolute values over the 1998 to 2020 period.

All spatial datasets were remapped to the grid of lowest spatial resolution (Table 1) using the bilinear method in the Climate Data Operators (CDO; Schulzweida. 2022). We excluded areas with bathymetry shallower than 5 m to circumvent dynamics related to intertidal zones. For analysis, coastal and offshore areas were defined as areas with bathymetry above and below 30 m, respectively. The Dogger Bank was considered in the offshore region.

The Chl-a in situ data was monthly averaged and monthly anomalies were calculated. We extracted the nearest grid point from the Helgoland Roads location in the remote sensing gridded data (hereby referred to as HRsat) and used monthly anomalies for comparison and validation of the GlobColour Chl-a dataset.

**Table 1: Description of spatial resolution and source of the parameters used in this study.**

| Variable | Spatial Resolution | CMS Product ID |
|---|---|---|
| Chl-a | 1 km | OCEANCOLOUR_ATL_CHL_L4_REP_OBSERVATIONS_009_098 |
| SST | 0.05° | SST_GLO_SST_L4_REP_OBSERVATIONS_010_024 |
| MLD | 0.111° × 0.067° | NWSHELF_MULTIYEAR_PHY_004_009 |


## 2.6 Evaluation of satellite derived Chl-*a* data

We compared the HRsat monthly anomalies time series with the in situ (HPLC) Chl-a monthly anomalies from the Helgoland Roads Time Series (HRTS) from 2004 to 2020. In this case, only the matching days were used for the calculation of the monthly anomalies, in both HRsat and HRTS. Our primary focus was on trends and monthly variability to assess the degree

of coherence between the in situ and remote sensing datasets. For the evaluation of trends, we applied the Mann-Kendal trend test (Kendall, 1975; Mann, 1945) and we made use of a boxplot for the variability assessment. The coefficient of correlation (r) and root mean squared error (RMSE) were computed to evaluate the goodness of fit between in situ and remote sensing



data. Differences in distribution between Chl-a in situ and remote sensing were verified by the two-sample Kolmogorov-Smirnov test.


## 2.7 Statistical Methods

As a pre-analysis, we calculated temporal mean and standard deviation (std) of the Chl-a anomalies. Using the mean and std, we computed the coefficient of variation (CV), calculated as $CV = \frac{mean}{std} \ x \ 100$, in % (Morel et al., 2010). Additionally, we examined linear trends in Chl-a anomaly fields. The significance of the linear trends was determined using a two-sided Wald test with t distribution. For more robust identification of significant positive and negative trends, we applied the Mann-Kendall trend test to the Chl-a anomalies. This was done on a pixel-by-pixel basis using the Python "pyMannkendall" library (Hussain et al., 2019). We calculated the Probability Density Function to investigate the changes occurring in the distribution of Chl-a anomalies.

We examined the relationship between the Chl-a anomaly fields with SST and MLD anomalies applying linear correlations in the direct anomaly fields and in 1 time step lagged Chl-a in relation to SST and MLD. For the analysis of dominant modes of Chl-a variability and covariability with SST and MLD, we used Maximum Covariance Analysis (MCA), a statistical technique that identifies prominent patterns of covariation (Bretherton et al. 1992) to maximize the covariability of associated parameters. MCA was designed to find patterns in two space-time data-sets that explain the maximum fraction of the covariance between them. This can provide insight into the physical processes leading to the spatial and temporal variations exhibited in the fields being analysed. Given the known significance of SST and MLD forcing in inducing chlorophyll changes (de Mello et al., 2022), this technique is particularly suited for our purpose. In essence, MCA extracts the singular vectors of the cross-covariance matrix of two fields, in order of importance. These singular vectors, also referred as structures or modes of variability, are extracted. When MCA was calculated using only one field, such as the Chl-a anomalies, we obtained the Empirical Orthogonal Functions (EOFS), which represented the leading modes of Chl-a variability. We utilized the Python package "xmca" (Rieger 2021) to apply EOFS and MCA. For the EOFS analysis, we used the monthly climatological Chl-a concentrations and the Chl-a anomalies, normalized during the analysis. For the MCA, the Chl-a, SST and MLD anomalies were employed.

## 3. Results

### 3.1 Evaluation of in situ HPLC and Remote Sensing Chlorophyll-a

Figure 2a illustrates the comparison of HRTS and HRsat Chl-a monthly anomalies time series. Both time series showed significant negative trends, evaluated by the Mann Kendall trend test. While satellite observations accurately reproduced the



intra- and inter- annual frequency, there was a discrepancy, with remote sensing tending to overestimate chlorophyll at low concentrations (Fig. 2b). In fact, satellite products tend to overestimate Chl-a values when they are less than 1 mg m$^{-3}$ (Alvera-Azcárate et al., 2021). When comparing the in situ HRTS and HRsat anomalies (Fig. 2a), we found a correlation coefficient (r) of 0.59 (p-value<0.05) and root mean squared error (RMSE) of 1.09 mg m$^{-3}$ (Fig. 2b). These values are in the range of values described in other works and considered acceptable for case-2 waters (Silva et al., 2021; Pramlall et al., 2023), in which the remote sensing product is defined by other constituents besides chlorophyll, such as coloured dissolved organic matter and non-algal particles (Doerffer and Schiller, 2007).

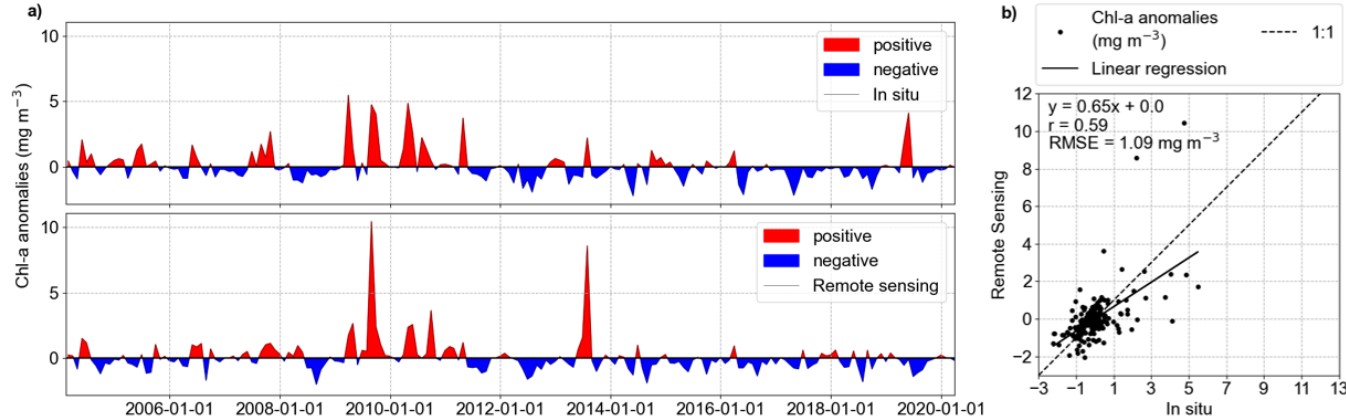

**Figure 2: Evaluation of GlobColour remote sensing surface Chl-a, compared to in situ data from Helgoland Roads. a) Comparison of in situ and remote sensing Chl-*a* anomalies, with red positive and blue negative anomalies. b) Scatter plot with linear correlation of the time series showed in a). Correlation coefficient is 0.59 and RMSE 1.09 mg/m$^3$.**

In the boxplot (Fig 3a), we observe that the in situ data has higher variability than the remote sensing data in April and May, months characterized by the phytoplankton spring bloom (Wiltshire et al., 2008). A rigorous assessment using the two-sample Kolmogorov-Smirnov test revealed no statistically significant differences between the anomaly distributions (p-value=0.83; Fig. 3b). The differences between in situ and remote sensing data were mitigated by using monthly mean anomalies. As a comparison, the evaluation using daily matchup Chl-a anomalies time series were r=0.35 and RMSE=2.64 mg m$^{-3}$.



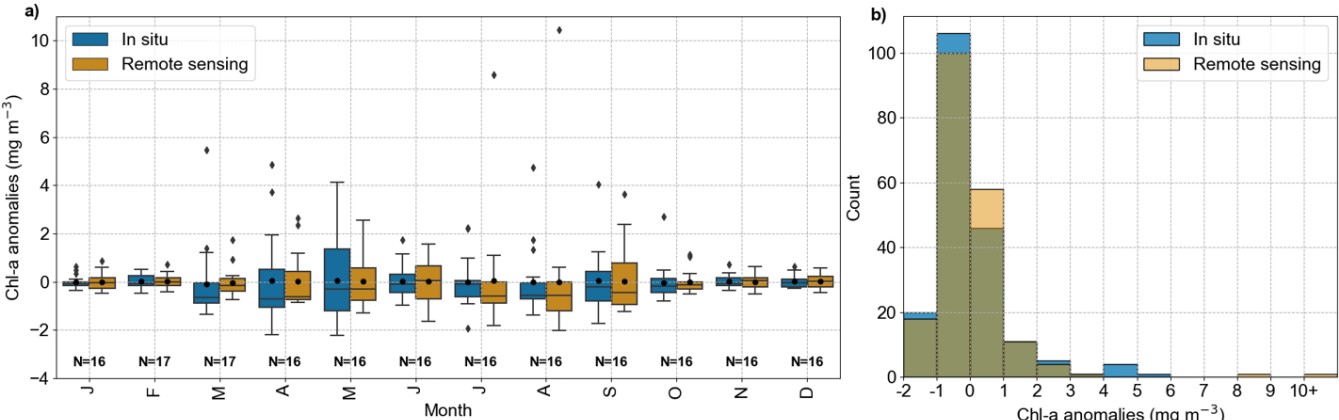

**Figure 3: a) Boxplots and b) distributions of remote sensing (orange) and *in-situ* (blue) monthly Chl-*a* anomalies.**

## 3.2 General findings:

The average of Chl-a (Fig. 4a) showed that higher concentrations (> 2 mg m$^{-3}$) were generally found near the coast, in areas with bathymetries less than approximately 30m. This excludes the shallow region of Dogger Bank (Fig 1). Areas with higher standard deviation (sd > 1 mg m$^{-3}$) were also found in shallow areas, with a depth less than 30 m. The increased variability in these shallow areas is primarily due to larger seasonal differences compared to the offshore waters (see Amorim and Wiltshire et al 2023 on temperature and seasonal variability comparisons for shallow and offshore sites). The standard deviation reflects the inter-pixel variability. Therefore, the coefficient of variation (CV) provided a measure of the spatial heterogeneity within the study area (Morel et al., 2010). The CV showed that 60% of the studied area is between 40 and 60%, a medium term between stable and large fluctuations. Two areas with larger CVs are in the north, around the Dogger Bank and the Danish coastal zone where there is a large bathymetry gradient; another area with high CV was found in the southern shallow waters of the Dutch coast, which is influenced by the water inflow from the English Channel. In the central region of the GB, we have identified significant negative linear trends with values ranging around from -0.01 to -0.03 (mg m$^{-3}$) per year. In contrast, the south-eastern corner of the study area, which is influenced by fresh water runoff from the river Elbe, exhibited positive significant linear trends (up to about 0.02 (mg m$^{-3}$) per year).



**Figure 4: The a) temporal mean and b) standard deviation of chlorophyll-a concentrations from January 1998 to December 2020.**
**Solid and dashed red lines represent 1 and 2 mg m⁻³, respectively, and the grey and black dashed lines are the 10 and 30 m isobaths.**
**c) Coefficient of variation, in percentage. d) Trends of Chl-a anomalies (mg m⁻³ per year). The shaded areas are significant (p-values**
**<0.05; two-sided Wald test with t distribution).**

Since 1962 there has been a notable SST increase in the North Sea, a trend that persists to the present (Amorim and Wiltshire et al., 2023). Specifically within the German Bight, the mean SST anomaly trend, as estimated by the locally weighted scatterplot smoothing method (LOWESS; Cheng et al., 2022) indicated an increase of 0.77°C from 1998 to 2020 (Fig. 5). This was further confirmed by the Mann-Kendall trend test, which showed a significant positive trend (p-value<0.001). However, when it comes to the averaged MLD, no significant trend was observed.



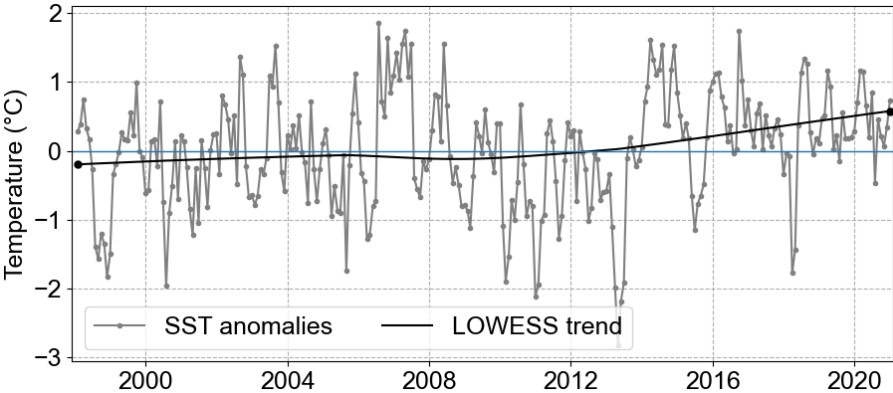

**Figure 5: SST anomalies averaged for the whole German Bight (grey line) and trend from 1998 to 2020 calculated by LOWESS (black line).**

## 3.3 Seasonal Chlorophyll-a surface concentration

The monthly climatological means (Fig. 6) were characterized by decreasing concentrations from coast towards offshore areas. The months of April and May exhibited elevated chlorophyll concentrations, a result that aligned with existing literature on the spring blooms of diatoms (Wiltshire and Manly 2004; Wiltshire et al 2010; Neumann et al., 2021). A slight increase was observed in August, indicative of the late summer/autumn bloom of dinoflagellates (Yang et al., 2021).



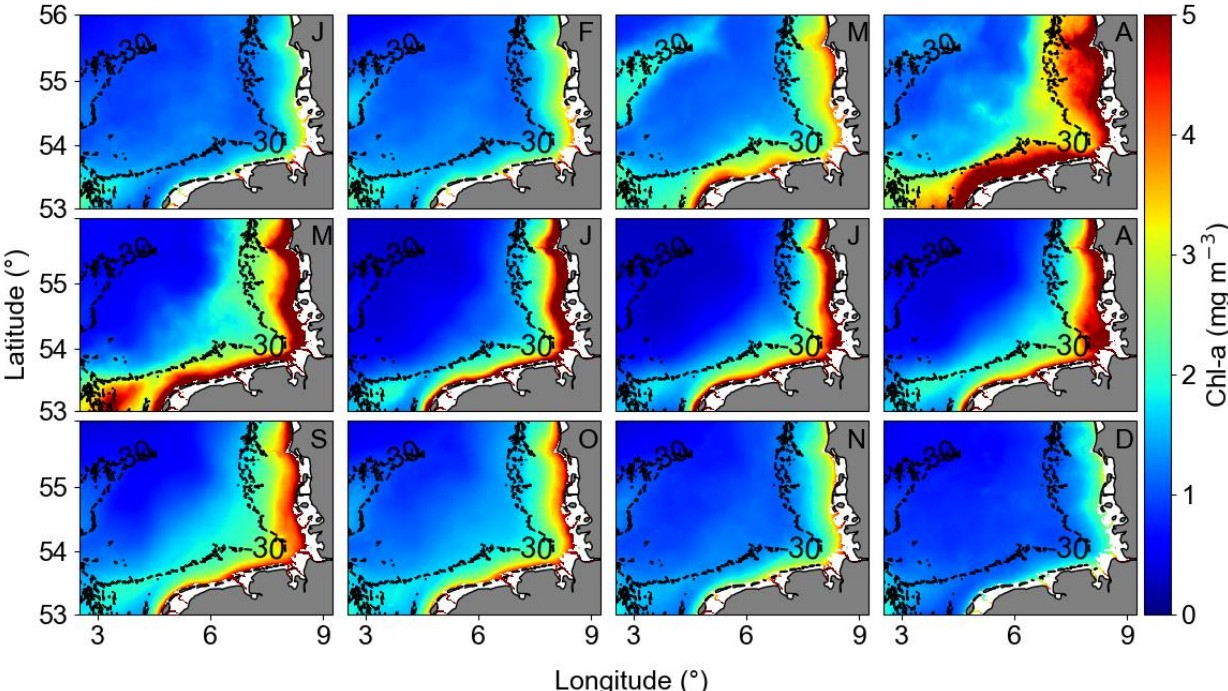

**Figure 6: Monthly-averaged GlobColour Chl-a, from January 1998 to December 2020 (letters on top right). The Chl-a monthly anomalies are calculated subtracting the monthly-averaged Chl-a from the absolute Chl-a data. It is possible to observe the intra-annual behaviour of Chl-a, with a positive gradient from open waters to coast, and the increase in Chl-a in April and August.**

Figure 7 illustrates the seasonal cycle of Chl-a remote sensing, spatial averaged for coastal and offshore areas, and HRTS in situ. Both coast and offshore were defined by a larger Chl-a peak in April, describing the phytoplankton spring bloom. Higher variability was observed in the coastal area, and a smaller peak, representing the late summer/autumn bloom, was observed in August. Values bellow 2 mg m$^{-3}$ were observed from November to February. The offshore area second peak was more perceptible compared to coastal areas, and it was observed in September/October. The summer Chl-a decrease was more accentuated in relation to the second peak in offshore areas, characterizing the period of lowest Chl-a, below 1 mg m$^{-3}$. The in situ HRTS acquired in the transitional zone of the German Bight, between coastal and offshore areas, aligned well with the spatial averages of Chl-a remote sensing, although the spatial averaged Chl-a remote sensing was overestimated during winter months, and the second bloom peak was delayed in offshore areas.



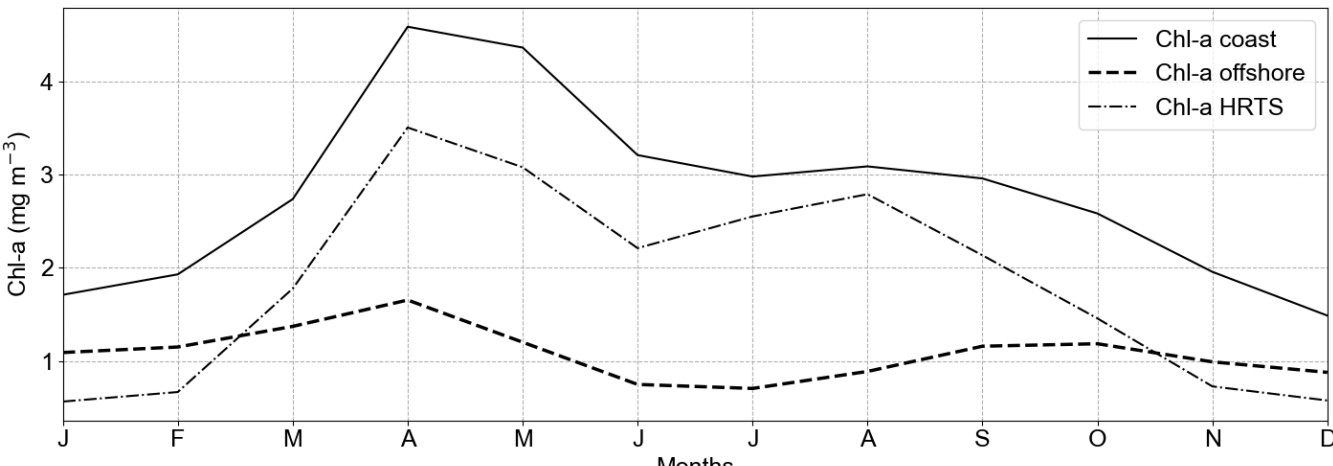

**Figure 7: Seasonal cycle of Chl-a averaged for areas above the isobath of 30m (coast), areas below 30m (offshore) and HRTS.**

The Empirical Orthogonal Functions (EOFS) applied to the climatological Chl-a values in Figure 6 revealed a dominant seasonal variation. This variation was represented by the first and second modes, which together explained 88% of the total variance in the annual cycle of Chl-a in the region (Fig. 8). For interpretation, we made use of the variability signals in the spatial modes/structures (EOF) defined by the red and blue colours (positive and negative, respectively) together with the signal of the associated principal component (PC). The first spatial mode (EOF1) accounted for 53% of the annual Chl-a

variability, exhibiting a positive signal across the entire German Bight. The first principal component (PC1) showed the two bloom signals, the spring bloom, in April, and the late-summer bloom, in September. Overall mode 1 described the maximum positive Chl-a variability during the spring bloom and, to a lesser extent, the late summer/autumn bloom (same signal in the spatial and PC temporal pattern), in contrast with winter and summer months, described by negative variability (opposite signals). The second spatial mode (EOF2) explained 35% of the Chl-a variability, dividing the GB in two regions based on the

variability between cold and warm months. The coastal/transition region was defined by maximum negative variability during the winter months and positive during summer months, according to the signals of EOF2 and PC2. Opposite, the offshore region showed negative variability during summer and positive variability during winter. This matched with the seasonal cycle of Chl-a shown in Figure 7, which represented the minimum Chl-a months in coastal areas during winter, but during summer in the offshore region.

Following EOFS mode 1 explained variability, we determined April and September as the most contrasting months for explaining the positive Chl-a variation in association with the environmental and biological/ecological drivers, and winter and summer as the negative variability months.



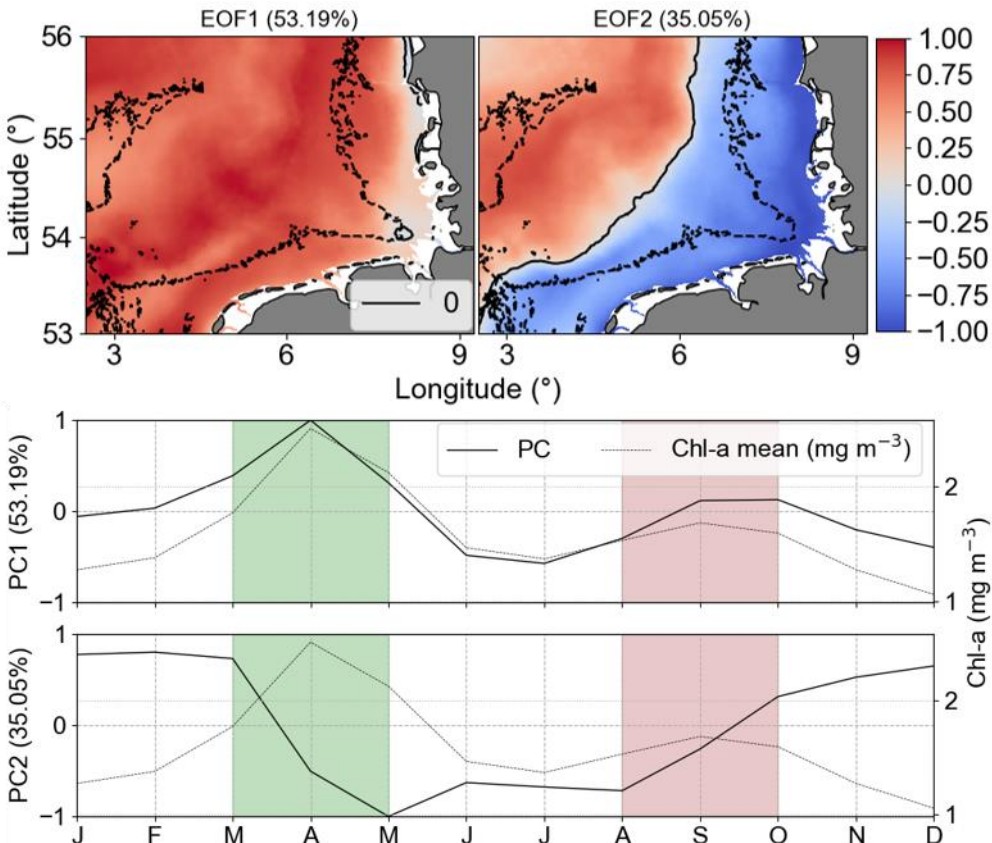

**Figure 8: First and second EOF and PC modes of monthly climatological means of Chl-a. Dashed thin line is the Chl-a spatially averaged at the study area. March-April (green shaded) and August-September (pink shaded).**

## 3.4 Month of maximum Chlorophyll-a concentration

96% of the German Bight area here analysed had a maximum climatological chlorophyll-a observed in March, April and May (20%,63% and 13%, respectively), representing the main months of primary production, and mainly related to diatom blooms (Wiltshire et al, 2008).

One interesting result for the remote sensing assessments was the month with maximum chlorophyll concentration (Fig. 9). Around Helgoland Roads position, August was, according to the remote sensing data, the month with maximum chlorophyll.

However, this is not consistent with the HRTS HPLC dataset. This might be an influence of the suspended matter dynamics (Fettweis et al., 2012) and/or the different pigmentation in phytoplankton species. Considering the two blooms occurring in the area, the spring bloom is characterized by dominant abundance of diatoms while the late summer/autumn bloom has also



high abundance of dinoflagellates, more prone to develop in stratified periods and with different pigment compilations (Shang et al., 2014; van Leeuwen et al., 2015). There is a defined difference between the reflectance spectra of diatoms and

dinoflagellates due to the distinct amounts of pigment types in each species (Shang et al., 2014) and such enhanced chlorophyll-a could be associated with cellular motility and the ability to regulate position in the water column, resulting in enhanced near-surface aggregation of flagellated cells (Franks, 1992). The Dogger Bank area was characterized by an early maximum in March, whereby could be that the shallower bathymetry allows the development of the spring bloom earlier due to light availability and enough nutrients for the growth of phytoplankton (Moll, 1997; Los et al., 2008).




**Figure 9: Month (colour bar from January to December) with maximum Chl-a by pixel from the monthly climatology means of Chl-a (Fig. 6; mg m⁻³). April (light grey) is the dominant month.**

**3.5 Chl-a distribution before and after 2009**

We examined the Chl-a anomalies spatial averages for the months of March, April and May over the 1998-2020 period to visually identify any potential increase or decrease of Chl-a, assessing interannual variability in coastal and offshore areas (Fig. 10). As the Mann Kendal trend test did not point any significant trends in the averaged Chl-a anomalies, we analysed the



changes in Chl-a anomalies distribution, splitting the time series based in the April peak observed in 2008 for both coastal and

offshore areas (Fig. 10). The peak in Chl-a anomalies in 2008 was related with a positive peak of North Atlantic Oscillation

index winter mean (NAO) (Fig. 10). In 2010, negative Chl-a anomalies peak was observed in April and May in the offshore

area and, in coastal area, in May, coinciding with an also negative peak of NAO. After 2010, a positive NAO winter trend was

observed. Considering these observations, we defined two periods to analyse Chl-a anomalies distribution: until 2009 and after

2009.

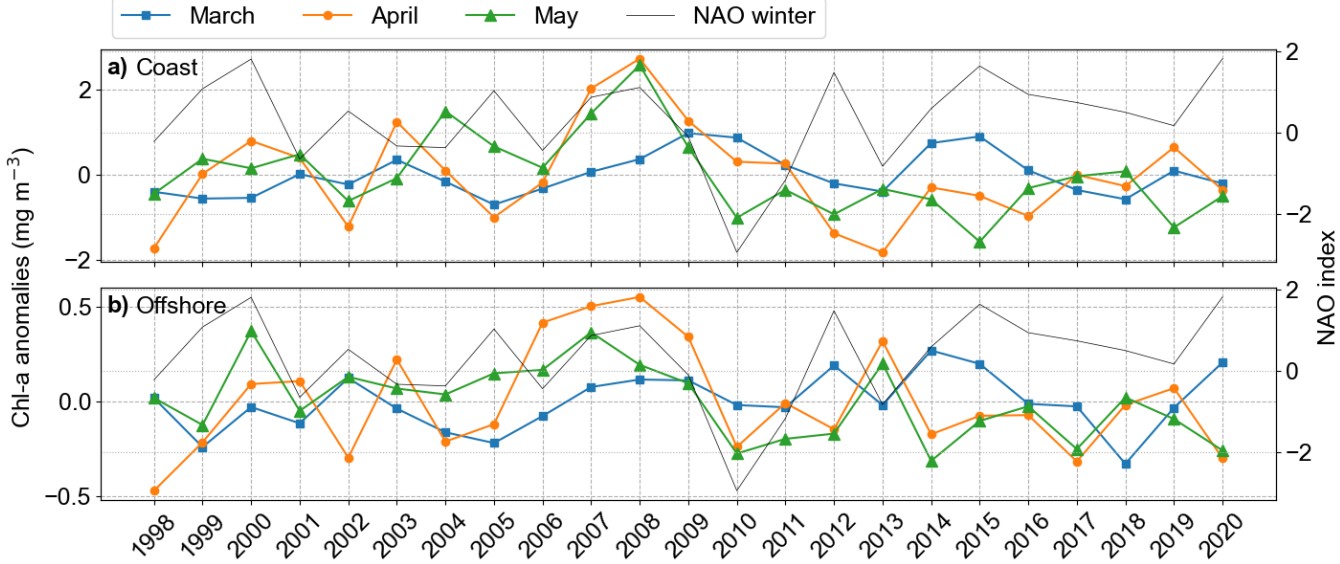

**Figure 10: Spatial averages of Chl-a anomalies in coastal waters (a) and offshore (b) for the months March, April and May. The NAO index winter mean (December, January, February) is the black thin line.**

We calculated the Probability Density Function of the two periods to investigate the changes occurring in the distribution of

Chl-a anomalies (Fig. 11). For the coastal area, March showed a shift from slightly normal to a bimodal distribution, with a

small negative bias related to the mean. The bimodal distribution was still dominated by Chl-a negative anomaly values but

with a lower second peak in positive values, indicating years with positive Chl-a anomalies. The offshore area distribution

described an increase in variance and increase of positive anomaly values. These results could be the response of earlier spring

blooms in the period 2010-2020 compared to the years before. April showed the highest variance change, decreasing from the

first period (1998-2009) to the second period (2010-2020). The decrease in positive anomaly values was evident and it can be

considered as part of the negative trend observed in the German Bight, together with the shift in May, moving completely from

positive to negative anomalies (Fig. 11).



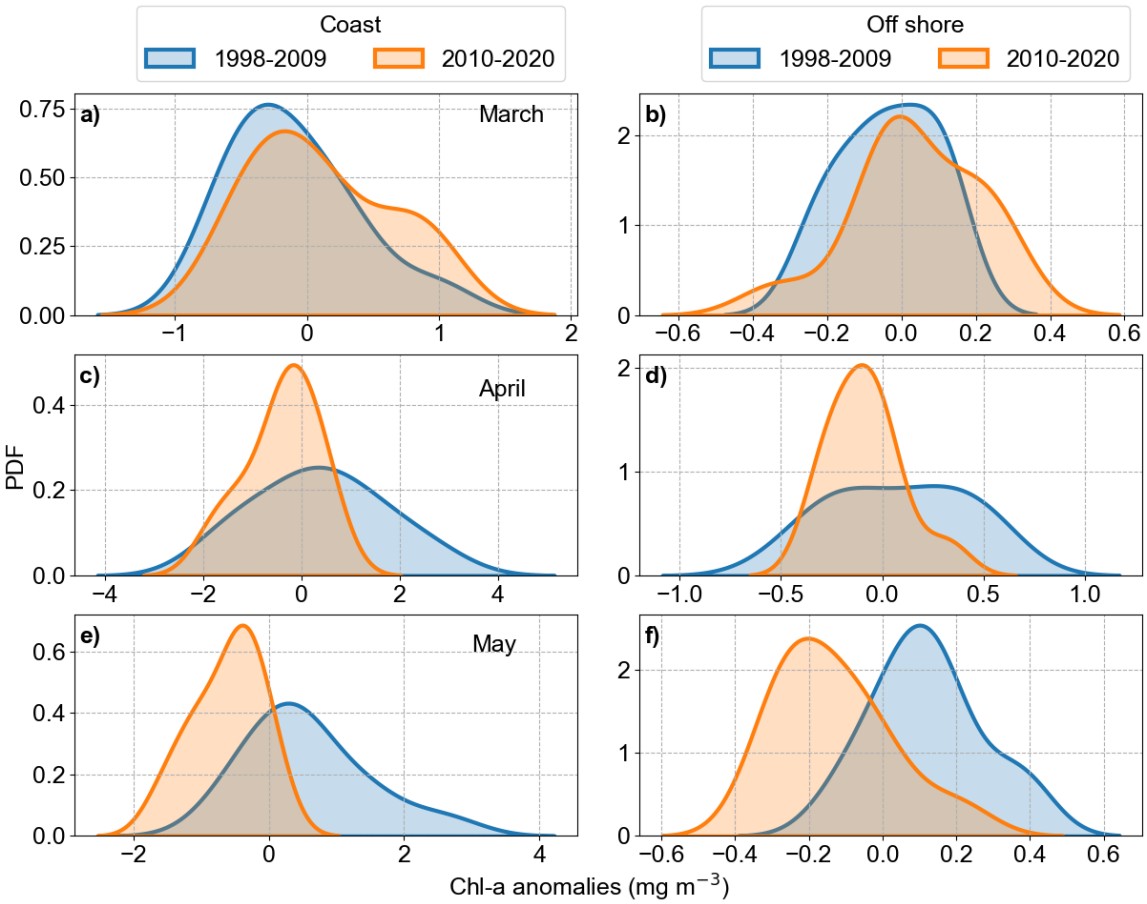

**Figure 11. Probability Density Function estimated from 1998-2009 (blue) and 2010-2020 (orange) for the Chl-a anomalies in March, April and May in the coastal and offshore areas of the German Bight.**

## 3.6 Chlorophyll-a Overall Trends

Figure 12 shows the trends of Chl-a anomalies in the GB. Mann-Kendall trend test showed that Chl-a anomalies significantly decreased in 31% of the analysed area (p<0.05). In the coastal area, mostly defined in the south-eastern corner of the study area, we found significant positive trends, covering 4% of the analysed area. In this region, there are available nutrients because of continued river input. Even with the turbid characteristic waters due to the river plumes influence, light availability is not a limitant (Fichez et al., 1992; Kerimoglu et al. 2017).



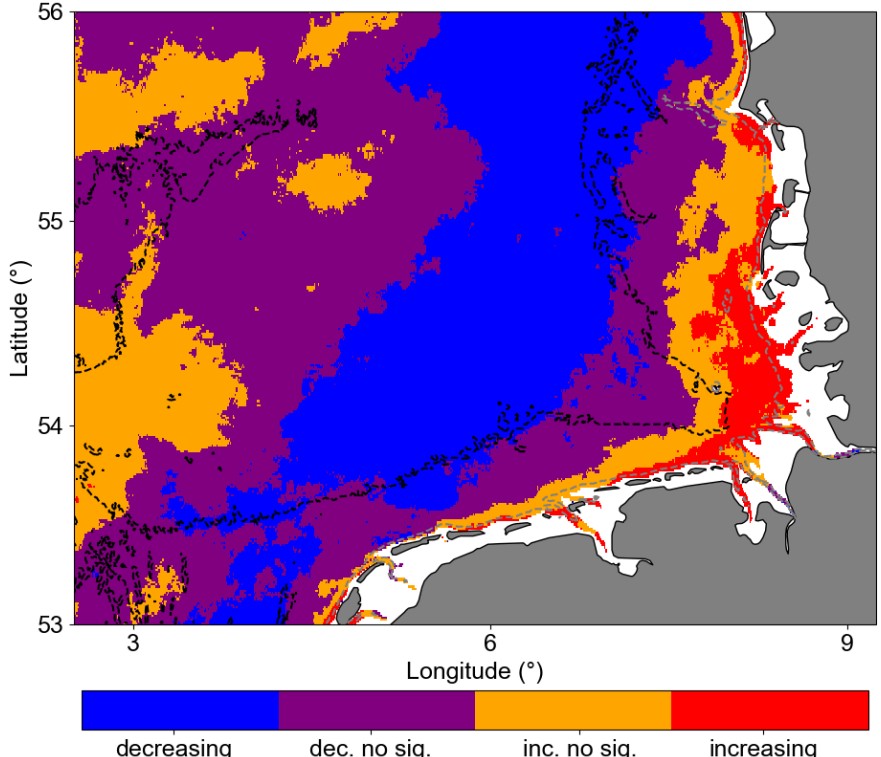

**345**

**Figure 12: Trend significance of Chla-a anomalies computed with Mann-Kendal trend test. Pixels with p-values <0.05 are considered significant. dec. no sig.= decreasing not significant, inc.no sig. = increasing not significant.**

Most of the central German Bight showed significantly negative trends. Associating the observed overall trends with the observed distribution changes, Amorim and Wiltshire et al (2023) examined the winter mean NAO index and computed a

positive trend. Also, the period after 2010 is characterized by long phases of a positive NAO index (Fig. 10, thin black line). Naturally, the positive NAO phase is associated with strong and frequent westerly (W) and southwesterly (SW) winds during winter and spring. In addition, also localized cyclonic systems over the British Isles can generate such winds (see Rubinetti et al., 2023). As expected, there is a positive trend within the 21st century in a frequency of W-SW winds (Rubinetti et al., 2023). Due to the Ekman transport, SW and W winds suppress the spreading of coastal waters from the south of the German Bight,

offshore, and intensify counter-clockwise wind-driven circulation in the German Bight (Schrum, 1997; Chegini et al., 2020). In addition, NAO in its positive phase characterizes strong Atlantic wind-driven inflow through the English Chanel, increasing mean temperatures in the North Sea (Pingree, 2005). In the spatial SST average (Fig. 13), we can see a tongue with North Atlantic water temperature characteristics, i.e. warmer than the characteristic German Bight surface water.  This means that negative and positive trends in Chl-a offshore and coast, respectively, can partly be explained by the wind pattern and in

particular by increasing frequency of W-SW winds during winter and spring, which leads to limited offshore spreading of rich nutrient coastal waters, and increases the warm Atlantic water inflow into the North Sea.



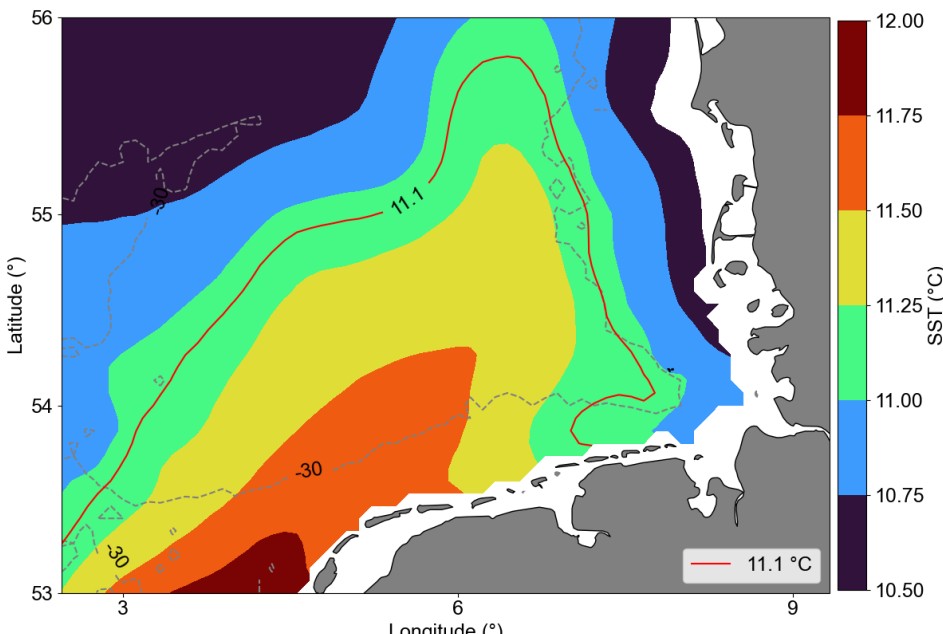

**Figure 13: Sea Surface Temperature time averaged for the study area. The red line marks the 11.1 °C isotherm.**


### 3.7 Dominant modes of non-seasonal Chlorophyll-a Variability

EOFS analysis was applied to examine the long-term non-seasonal variability in more detail. Recent studies have used this type of analysis on chlorophyll remote sensing and model simulation spatial data to detect the influence of environmental/oceanographic processes on phytoplankton biomass time/space variability (Daewel and Schrum, 2017; Alvera-Azcárate et al, 2021). Results of our EOFS analyses showed that the first four modes accounted for 45% of Chl-a non-seasonal variability in the study area. The percent variance explained by each mode was 19.02% (mode 1), 11.79% (mode 2), 8.36% (mode 3) and 5.54% (mode 4). Figure 14 shows the spatial patterns for the EOFS modes 1–4 which are associated with dominant long-term non-seasonal features, since seasonal frequency cycles have been removed by the climatological monthly means subtraction procedure. The normalized amplitude time series (principal component, PC) corresponding to the spatial patterns (EOF) and the monthly means are shown in Figure 15a-d and Figure 15e-h, respectively. The PCs represent the time evolution of all pixels in the corresponding mode spatial pattern. If a pixel in the spatial pattern and its associated temporal amplitude have the same sign, it means a positive chlorophyll deviation for that pixel at that time, in relation to the zero value in the spatial map. Conversely, when pixels in the spatial pattern and associated temporal amplitude show opposite signs, it means a negative deviation from zero. Therefore, pixels that show similar signs and values in the spatial pattern (Fig. 14) have similar behaviour in time and represent coherent features (Garcia and Garcia., 2008).



In terms of the spatial variability of Chl-a, we show that there were distinct spatial and temporal variability in the German Bight concerning the modes of variability.

EOF1 showed the same signal for the whole German Bight i.e. a decreasing variability from coast to offshore regions and, as indicate by the PC1, it is mostly related with interannual variability during spring blooms and positive peaks after the late

summer/autumn bloom. EOF2 split the area into offshore (northwest region) and transitional/coastal waters from southwest to northeast. EOF3 was characterized by a positive signal in the southwest region, possibly related with the English Channel inflow, bringing warmer waters to the German Bight. EOF4 can probably be related to stratification for a relatively long time mediated by the wind forcing and river forcing associated with Elbe and Weser fresh water discharges (Chegini et al., 2020). It is important to point that the modes or structures of variability contain information about the variability of the dataset that is

not necessarily related directly to physical features. The interpretation of the EOFS modes just aims to relate the data modes with physics (Olita et al., 2011). The temporal amplitudes associated with the spatial patterns are also object of physical interpretation. To facilitate the interpretation of the PCs, we applied the spectral analysis information (Fig. 15i-l ).

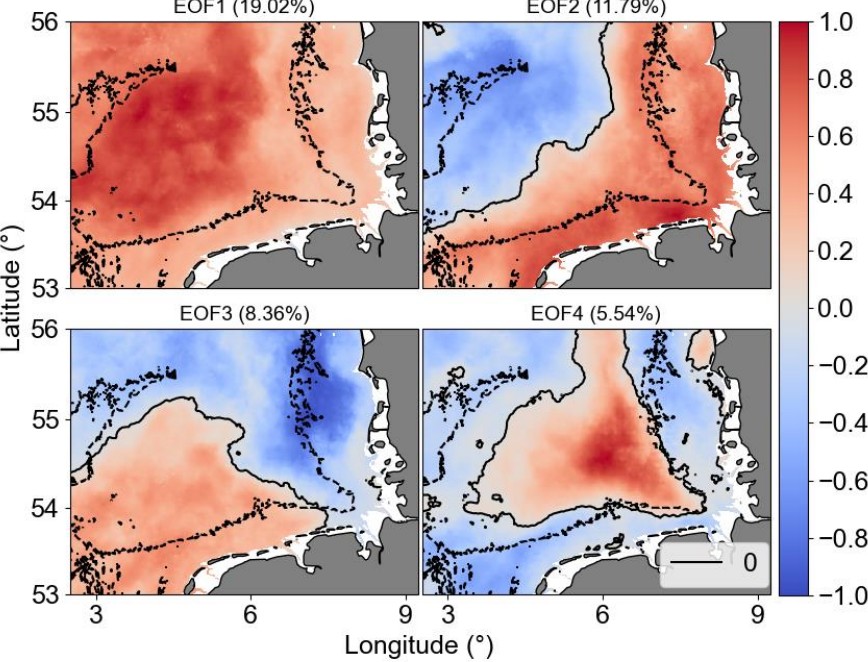

**Figure 14: First four Chl-a anomalies EOF and PC modes. In EOF fields, black solid line is the 0 amplitude contour and black**
**dashed lines are the isobaths of 30m. The amplitude values shown are all max-normalized. Explained variability percentage for each mode are depicted inside parenthesis.**



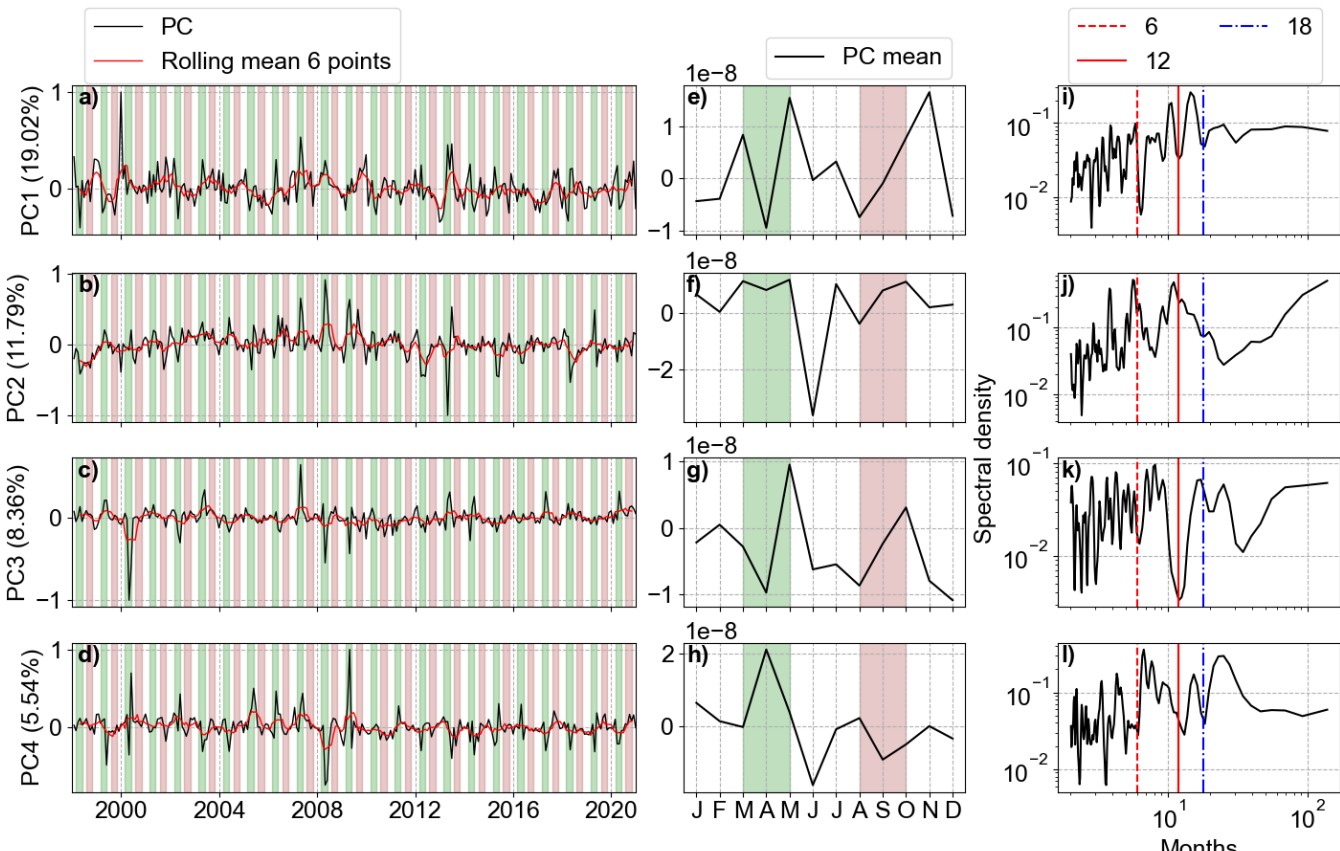

**Figure 15: First four Chl-a anomalies PC. a-d) the PC amplitudes (black) with 6 point rolling means (red solid). e-h) PCs monthly means. i-l) Spectral analysis of PCs time-series. The amplitude values are max-normalized. Explained variability percentages are depicted inside parenthesis. March-April (green shaded) and August-September (pink shaded).**

The spectral analysis of the Chl-a anomalies PC1 identified the highest peak of energy around 15 months and another high peak at 11 months, related to interannual variability, mostly at the spring bloom April month, while PC2 has peaks at 4, 6 and 12 months, linked to the intra annual variability. In the averaged PCs, we can see that PC1 accounted for the variability of the two blooms, while PC2 was related with the decrease of Chl-a during summer months, happening mainly in deeper areas due to nutrients depletion during the spring blooms and the zooplankton grazing. PC3 and PC4 together accounted for approximately 14% of the variability and seemed to be characterized by extreme values occurring in sporadic periods due to the English Channel and Elbe river inflows, as observed in the EOFs.

**3.8 Chlorophyll-a, Temperature and Mixed Layer Depth relationships**



Linear correlations revealed significant but not strong relationships between Chl-a anomalies variability and SST or MLD anomalies at the study area (Fig. 16). For Chl-a and SST correlations, the southern coastal area (along Germany and

Netherlands border) was characterized by significant positive correlation, while a patch of negative correlation was observed close to the Dogger Bank area. For Chl-a and MLD, most offshore areas showed significant positive correlations, indicating that positive MLD anomalies were correlated to positive Chl-a anomalies. The correlations between Chl-a and MLD close to the coast and around isobaths of 30 m were negative. Lagged correlations between Chl-a and the two parameters (SST and MLD) did not provide higher correlations, indicating that the Chl-a variability possibly responds in time scales shorter than

monthly periods or longer time scales. Focusing in the areas with significant correlation in Figure 16, Chl-a and SST anomalies lagged correlations were negative around the Dogger Bank, where bathymetry changes from around 50 m to shallower than 30 m in the bank area. The southern coastal area of the German Bight is described by positive correlations, a result not aligned with the Chl-a overall trends, indicating an indirect effect of temperature on Chl-a. The Chl-a and MLD correlations are positive in most of the deeper parts of the GB, where higher Chl-a values were found and connected with deeper MLD. In most of the

areas shallower than 30 m, Chl-a and MLD anomalies were negatively correlated.



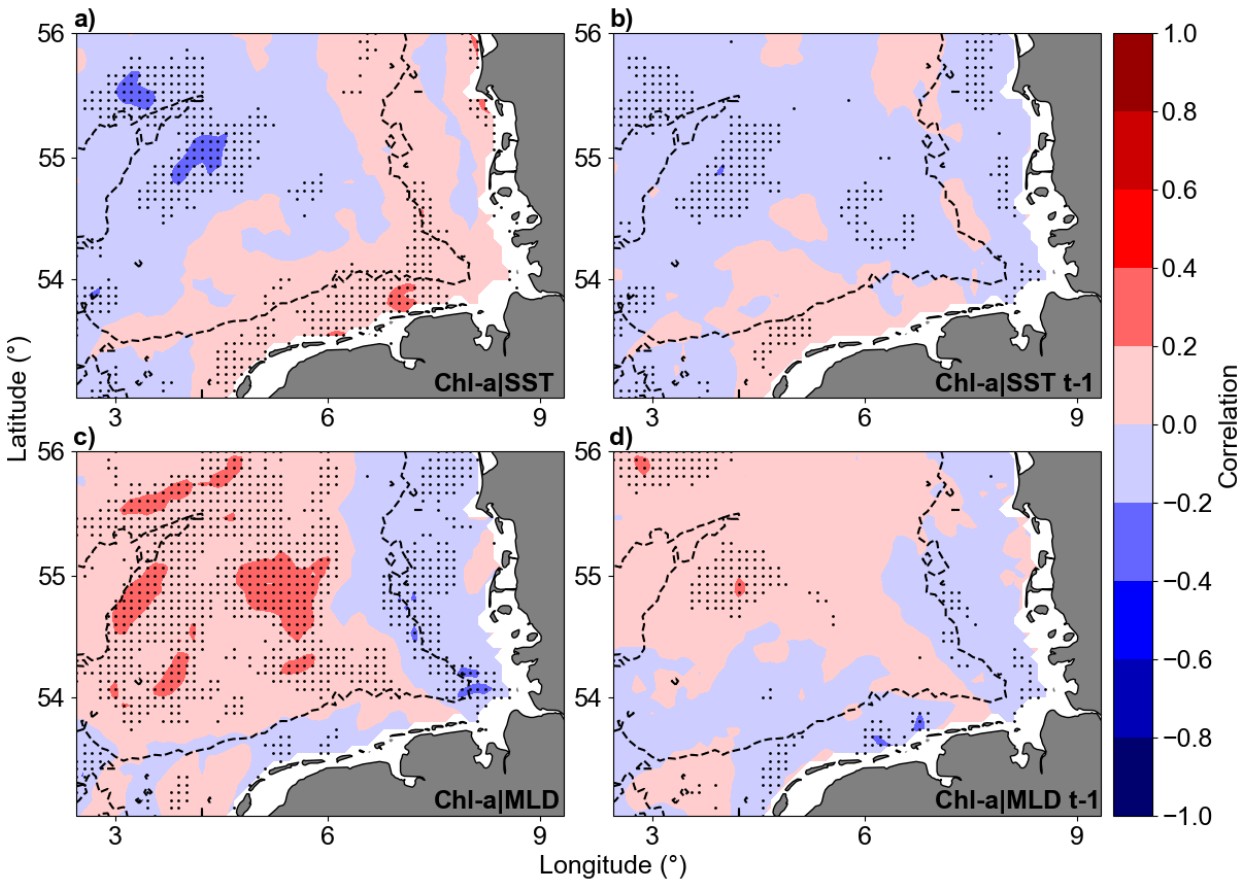

**Figure 16: Correlation maps of Chl-a and SST (a, b), and Chl-a and MLD (c, d). b) and d) lagged correlations with Chl-a lagged one month. Warm colours are positive and cool colours are negative correlations. The dotted areas are significant (p-values <0.05; two-sided Wald test with t distribution).**

Maximum Covariance Analysis (MCA) was carried out on Chl-a anomalies and SST and MLD anomalies to identify spatial patterns. With this we hoped to explain as much as possible of the mean-squared temporal covariance between the two fields (Bretherton et al., 1992). MCA produces two sets of singular vectors along with a set of singular values. The relevant property of these singular vectors is that they maximize covariance (Von Storch and Zwiers, 2001; Martínez-López and Zavala-Hidalgo, 2009). The correlation coefficient between the "same mode" principal components of the two fields quantifies the strength of the coupled maximum covariance described by that mode (Martínez-López and Zavala-Hidalgo, 2009). This possibly tells us about how strong the processes are directly connected or if there are other indirect processes involved, and about the time scale coherence between the variables (Fukutome at al., 2003; Rieger et al., 2021)



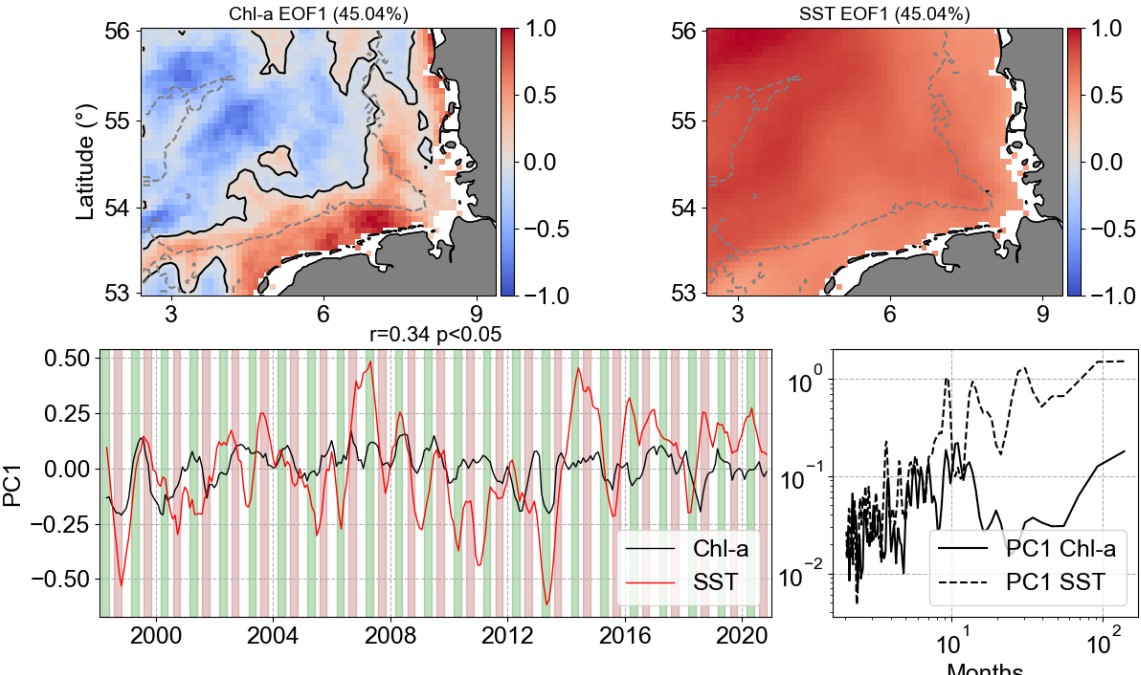


**Figure 17: MCA results applied to Chl-a and SST anomalies. PC1 is shown as rolling mean of 6 points.**

The MCA Chl-a|SST anomalies first mode (Fig.17) presented positive anomalies covering the whole German Bight for SST, while the Chl-a showed negative anomalies in the deeper offshore area, far from the coast, and positive anomalies in the depths

around 30 m and shallower. The largest negative Chl-a anomalies in mode 1 were in areas surrounding the Dogger Bank, and the positive ones in the coastal southern part of the German Bight. This pattern was already visible in the Chl-a|SST anomalies correlation presented in Figure 16. The first mode explained 45% of the covariance between Chl-a anomalies and SST anomalies, the non-seasonal variability. The result of MCA mode 1 was more or less what was observed in the spatial Chl-a correlation, with positive correlation in offshore waters and negative correlation in the coastal areas. In the offshore region,

we assume this is the role played by the critical depth theory (Sverdrup, 1953; Tian et al., 2011) and the weakening of turbulence after winter (Wiltshire et al., 2015). The PC1 showed a significant weak correlation of 0.34, meaning weak coupling between the two PCs. The spectral analysis of PC1 did not show connected peaks, indicating that intra-annual variability in Chl-a covariates with lower frequencies of SST. The mode 2 of Chl-a|SST anomalies MCA explained much less, only 9.2% (not shown).



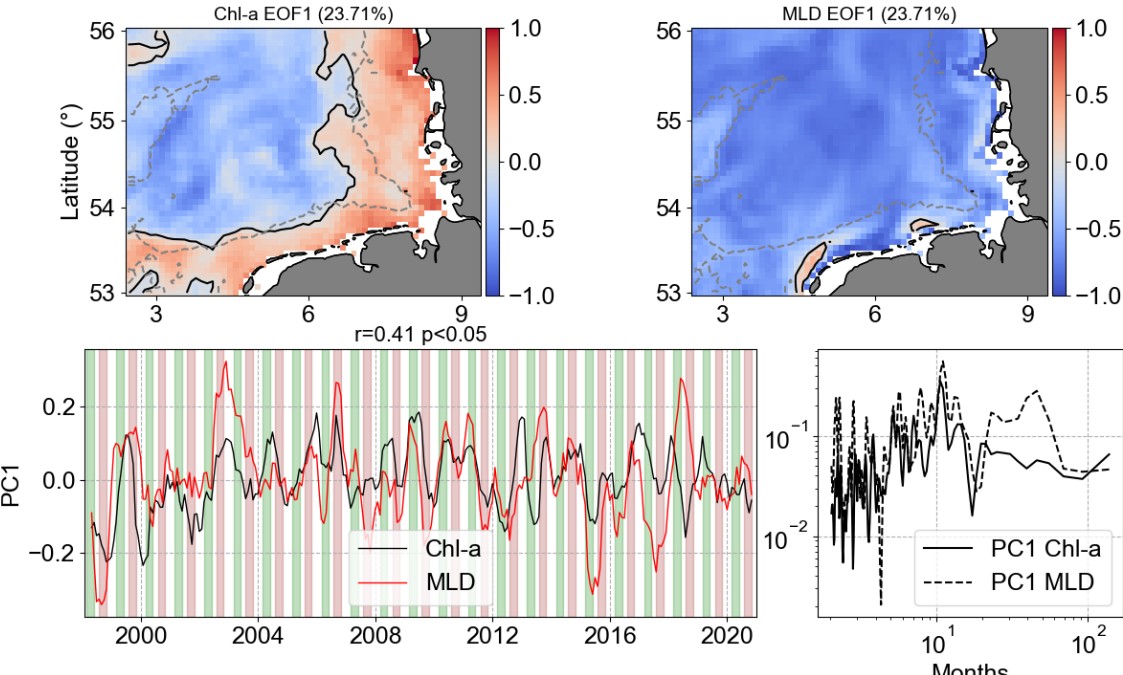


**Figure 18: MCA applied to Chl-a and MLD anomalies. PC1 is shown as rolling mean of 6 points.**

The MCA mode 1 of MLD anomalies had the same signal in the majority of the analysed spatial domain, but Chl-a mode 1 had a clear separation between coast and offshore. MLD and SST work in opposite ways for these two regions. Even though mode 1 of Chl-a and MLD anomalies accounted for approximately 23% of covariance, less than the Chl-a/SST, the Chl-a/MLD

PC1 spectral analysis showed slightly higher coherence, with non-seasonal changes in MLD affecting Chl-a in the same temporal scale.

## 4. Discussion and Summary

The use of the Copernicus GlobColour chlorophyll-a surface concentration allowed a comprehensive analysis of Chl-a long

term trends and variability (23 years) in the German Bight. The evaluation of the GlobColour remote sensing dataset using a long HPLC Chl-a time series from the Helgoland Roads showed good agreement in trends and variability. The seasonal intra-annual variability in coastal and offshore areas of the GB was defined by two Chl-a peaks, characterizing the spring and late summer/autumn phytoplankton blooms. Higher variability in coastal waters was observed between winter and bloom periods, while in offshore areas, the higher variability was between summer and the bloom periods. The distribution of coastal and

offshore area averaged Chl-a, before and after 2009, was characterized by clear changes in April and May Chl-a anomalies, with decrease of variance and the distribution peaks moving to negative values. Following the distribution changes, the overall



trends in the German Bight were described by a large area in the centre of the GB with significant negative trends, with only a limited area close to the Elbe river influence showing significant positive trends. The dominant modes of the non-seasonal Chl-a variability defined spatial and temporal components, associated to interannual variability of Chl-a during the bloom

periods and a distinction between coastal/transitional and offshore areas. The English Channel and river inflows accounted for a small fraction of the explained Chl-a variance. The covariability of Chl-a anomalies with SST and MLD anomalies, assessed by Maximum Covariance Analysis, showed higher covariability between Chl-a and SST, but the spectral analysis and the lower PC correlation indicated distinct time scales of variability. In the case of Chl-a and MLD covariability, the value was almost half of the one observed for SST, but occurring in the same time scales. The low linear correlation between Chl-a and

SST could mean that there are indirect effects caused by temperature changes The direct positive influence of increasing temperature is limited and can be outweighed by negative indirect effects and other factors, such as nutrient availability.

It is important to point that the changes and variability in Chl-a cannot be assumed to happen only due SST or MLD, but as a combination of factors that can compensate or amplify each other (Xu et al., 2020). The relationship between SST and MLD together with the availability of light, nutrients and turbidity controls mostly of the primary production variability in the

German Bight. Changes in ocean surface temperature are associated with changes in other variables, including biological variables such as Chl-a, primary productivity, species physiological responses and species distributions (Dunstan et al 2018, SST and Chl-a). Potentially, warming can either affect directly and indirectly the Chl-a variability. Higher temperatures can alter the species physiological responses and species distributions (Dunstan et al., 2018). Wind patterns and fresh water discharge are impacted by climate change, and phytoplankton predation are enhanced by the increase in temperature due to

accelerated metabolism of zooplankton. The mechanism of influence becomes intricate as temperature modifies the physiology of species, species composition, river runoff, and other factors (van Beusekom and Diel-Christiansen, 2009; Capuzzo et al, 2018; Dunstan et al., 2018). Although there is a direct positive influence of increasing temperature, it is limited and can be outweighed by negative indirect effects and other factors, such as nutrient availability.

Balkoni et al (in prep.) estimated nutrients decadal changes in the German Bight and the results point out that there is a decrease

in nutrients. Capuzzo et al (2018) pointed that observed decrease in primary production in the North Sea from 1988 to 2013 are attributed to increasing temperature and decrease in nutrients, corroborated by Desmit et al., (2020) analysis in the southern North Sea. van Beusekom and Diel-Christiansen (2009) identified that higher temperatures favours zooplankton growth and grazing during the spring blooms, decreasing the intensity of phytoplankton growth. Alvera-Azcárate et al. (2021) observed decreasing trends and slight increase of Chl-a in the last 12 years for the whole North Sea, together with early onset of spring

blooms and positive trends in bloom duration, indicating the high variability and heterogeneity in the different regions of the North Sea. A combination of factors is the most likely scenario under climate change for the primary production in the German Bight, but it is important to assess the individual consequences to better understanding the whole.



**Conclusions**

The utilization of GlobColour Chlorophyll-a surface concentration enabled the identification of the primary modes of
variability in the German Bight. Overall, the comparison between remote sensing and in situ data demonstrated consistent
results in evaluating Chl-a surface variability. Specifically, when comparing the in situ HRTS and remote sensing monthly
anomalies, a correlation coefficient (r) of 0.59 and a root mean squared error (RMSE) of 1.09 mg m$^{-3}$ were determined. The
analysis of surface chlorophyll-a concentration trends and variability was conducted in conjunction with anomalies in sea
surface temperature and mixed layer depth. Spatial Chl-a trends reveal a significant decrease in concentration in the central
region of the German Bight over the past 23 years. In contrast, the near-coastal zone influenced by the Elbe River discharge
exhibits a notable increase in chlorophyll-a. Concurrently, a positive temperature trend is observed throughout the German
Bight. It can be concluded that its direct positive influence on chlorophyll-a concentration is limited and can be outweighed
by negative indirect effects and other factors, such as nutrient availability and wind conditions. Over the last 23 years, a positive
trend in the frequency of south-westerly and westerly winds during the winter/spring season has been observed, attributed to
a prolonged positive NAO phase during the considered period. Such winds prevent the offshore spreading of nutrient-rich
coastal waters from the southern German Bight and enhance the warm Atlantic water inflow into the central North Sea. This,
in part, can explain the contrasting Chl-a trends in offshore and inshore zones. The EOFS analyses of the chlorophyll-a
concentration showed the most variability in the intra-annual blooms (first mode) and the decrease in Chl-a during summer in
offshore areas and in winter months on the coast (mode 2), totalling around 88% of seasonal explained variance. For the Chl-
a non-seasonal EOFS results, the first mode (19%) was defined by inter-annual variability, with a peak of energy in the
spectrum at 11-months and 15-months cycle frequencies. This can be explained by the negative inter-annual variability in the
spring blooms and positive variability after late summer/autumn blooms. The first four modes explained 45% of variability,
defined by the division into offshore and transitional/coastal waters and intra-annual variability, the English Channel inflow,
and the presence of fresh water stratification. The MCA analysis showed higher covariability between Chl-a and SST
anomalies than Chl-a and MLD anomalies. The first mode of Chl-a|SST was represented by 45%, and the first mode of Chl-
a|MLD by 23%. For next steps, we suggest the analysis of nutrients in the German bight, relating possible changes with winds
and stratification. A combination of remote sensing Chl-a coupled with vertical in situ data would give even more insight about
the 3D variability of Chl-a in the German Bight.

**Data availability**

Data used in this paper are available in https://data.marine.copernicus.eu through the Copernicus Marine Service and in
PANGAEA Data Publisher for Earth & Environmental Science. The NAO Index Data was provided by the Climate Analysis
Section, NCAR, Boulder, USA, Hurrell (2003). Updated regularly. Accessed 11 November 2023.

**Author contribution**



FA conceptualized the study, processed the data, and carried out all data analyses. FA wrote the original paper with contributions from AB, VS, and KW. KW supervised the study. All authors reviewed and edited the final paper.

**Competing interests**

The contact author has declared that none of the authors has any competing interest.

**Acknowledgements**

This study has been conducted using E.U. Copernicus Marine Service Information. We acknowledge the immense effort of the Aade crew to collect the Helgoland Roads data and the people who carried out the chemical analyses.

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
