# Peer review of "Analyses of sea surface chlorophyll-a trends and variability from 1998 to 2020, German Bight, North Sea."

_EGUsphere, 2024_

## Referee Comment (RC1)

Review of ''Analyses of sea surface Chlorophyll-a trends and variability in a period of rapid Climate change, German Bight, North Sea''.

In this work, the authors provide a comprehensive analysis of sea surface chlorophyll-a trends and variability in the German Bight, a coastal area in the North Sea, using satellite remote sensing data and in situ measurements. The paper aims to understand the relationship between chlorophyll-a, sea surface temperature, and mixed layer depth. The paper presents some interesting and novel findings, such as the significant positive trend of chlorophyll-a near the Elbe estuary and the negative trend in most of the central German Bight, the changes in the distribution of chlorophyll-a anomalies before and after 2009, and the contrasting modes of co-variability between chlorophyll-a and sea surface temperature or mixed layer depth in coastal and offshore areas. Overall, the paper is well-written and structured, but I think some of the figures could be improved before publication. I have some major/minor comments and suggestions, but I could not see any scientific flaws, and think the manuscript is a good addition to the field.

1. In this MS, the authors have used several statistical methods such as EOF, MCA, linear correlation, trend analysis, probability density function, and different types of tests such as the Mann-Kendall trend test, Kolmogorov-Smirnov test, and two-sided Wald test. However, the authors do not examine/illustrate these statistical techniques in detail. For example, what is the LOWESS trend, and how can it be estimated? What is the two-tailed Wald test? Is it different from the t-test? Could you please add more details about this test? What is the difference between this test and the Mann-Kendal trend test? Also, I strongly recommend using the modified Mann-Kendal test (Hamed and Ramachandra Rao, 1998), which takes into account the serial correlation between observations. The authors should also provide more details and justifications for some methodological choices, such as the definition of coastal and offshore regions, the criteria for significance tests and confidence levels, and the number of modes used for the EOF and MCA analyses.
   https://doi.org/10.1016/S0022-1694(97)00125-X

2. The MS is overloaded with content, analyses, and details, which can be reduced in some places for better understanding and easier to follow at times. For example, figure 4D and figure 12 are identical because the authors have already superimposed the significant and non-significant regions in Figure 4D. In addition, the entire section (Section 3.6) in the description of Figure 4D can be moved to the main body of Section 3.2 so as not to interrupt the story. In another example, from Figure 6 and Figure 7 and their description, the authors came to the same conclusion that the highest chlorophyll concentrations are found in the coastal region in April and May. And so on ...
   I strongly recommend adding a file of supplementary material that includes these figures (6 and 12) and others that are not discussed in detail in the main body of the MS (e.g., fig. 13).

3. The MS does not provide a clear explanation for the choice of 2009 as the breakpoint for the analysis of chlorophyll-a anomalies distribution. It seems that this year was selected based on the peak of chlorophyll-a anomalies observed in 2008, but the paper does not discuss the potential causes or implications of this peak. It would be helpful to provide

more justification and context for this choice and to explore the sensitivity of the results to different breakpoints. To detect the abrupt change in chlorophyll-a concentrations, I highly recommend using the Pettitt homogeneity test (Pettitt, 1979).
https://doi.org/10.2307/2346729

4. Some parts of the paper are a repetition of the others, for example, Figure 8b does not bring any new results than those in Figure 7. Also, I wonder why the authors estimated the seasonal cycle of each principal component at the seasonal (Figure 8b) and interannual scale (Figure 15 E, F, G, and H) although it is supposed to use the PCs to look at variability during the whole study period. In my opinion, Figure 8 does not provide any new results and can be part of the supplementary material. In particular, the authors have already applied the EOF to the Chl-a anomalies (Figure 14 and Figure 15). Also, all spectral analyses applied to each principal component (Figure 15 I, J, K, and L) could be removed and applied the spectral analyses to the original data (Chl-a).

5. The authors mention "a period of rapid climate change" in the MS title. It is not clear to me whether the authors consider the whole study period as a rapid climate change or whether they defined this period in their MS using a specific test. Please add more details on this point or support it with a reference in MS or change the title.

6. Objectives: In Lines 80-85 the five main objectives of the study are stated. For me, objectives (ii), (iii), and (iv) seem to be identical to the main objective (line 75). I would suggest rephrasing/rewriting the main goals concisely and clearly. I would also suggest that the authors put these in the final section of the paper when summarizing their findings in the conclusion. What is the difference between objectives (ii) and (iv)?

7. Lines 233-237: In this section, more details on the SST time series in Figure 5 are needed, e.g. which year has the highest and lowest SST anomalies and variability. In addition, the SST trend values obtained should be compared with previous studies in the same region to highlight differences and similarities. Furthermore, I suggest creating the spatial trend maps of SST. This will give the reader a clear picture of the spatial and temporal variability of SST trends in different locations of the study area, which can be compared to the chlorophyll trend map.
https://doi.org/10.3389/fmars.2023.1258117
https://doi.org/10.5194/nhess-22-1683-2022

**Other comments**

- Figure 2: For the comparison, it would be better to draw a two-line time series in one panel instead of drawing the positive and negative anomaly for each one, which makes the comparison unclear.

- Figures 15,17, and 18, Shaded regions make these figures unclear, I suggest removing these shaded regions.

- As far as I know, it would be better to limit the acronyms in the abstract and introduce them in the text (from the introductory chapter onwards).

- The abstract is very long and contains a more general and longer sentence, which can be shortened or moved to another section (e.g., introduction). For example, a sentence starts in line 12 and ends in line 15. The same for the next one (lines 15-18). Please try to shorten the abstract to be concise and focus on the most interesting results, of which there are many in your MS.

- Line 13: Please use "comparing with the in-situ data" instead of "comparing with the Helgoland Roads Chl-a in situ data".

- Line 19, "A significant long-term positive trend was observed close to the Elbe estuary and adjacent area". The trend of what?

- Please indicate the source of the bathymetry data used in Figure 1.

- Line 96: Please add the position of the Elbe estuary in Figure 1.

- Line 100: please use "flow" instead of "inserted"

- Line 114: Please provide the doi and a reference to the data, if possible, instead of using the general link of CMEMS and the product name. Especially, the same link has been repeated in line 120 and line 125. Also, I suggest removing Table 1.

- Line 120: which products are used for SST and MLD? It is not clear to me. Please add more details about these products.

- Line 163 ''the two-sample Kolmogorov Smirnov test.'' please add the reference for this test.

- Line 189: I suggest removing the acronym HPLC from title 3.1. I understand that it was used previously and refers to "high performance liquid chromatography" but should not be used in the title.

- Line 190: "Both time series showed significant negative trends". Please add the values of these trends.

- Lines 214-226: please refer to fig4b, fig4c, and fig4d in this section.

- Line 229: In the caption of Figure 4, I think the authors should use the spatial mean instead of the temporal mean. Or they can use spatial climatological means.

- Line 236: ''However, when it comes to the averaged MLD, no significant trend was observed." On what basis do the authors come to this conclusion? Do they estimate the trend of temperature at MLD?

- Line 290: I suggest starting the sentence with something else instead of the number.

- Lines 402-408: What if the authors apply spectral analysis to the original data? Do they expect to get the same results?

- Figure 16; please use an appropriate range for the color bar, say between -0.4 and 0.4. It is not clear how the trends are significant in some regions and not significant in others, while both have the same trend values. Have you tried testing these correlations with different time lags and not just one month?

- Please move lines 432-439 to the methodology section.

- The MS does not provide a clear link between the observed chlorophyll-a trends and variability and the broader implications for the marine ecosystem and biogeochemical cycles in the German Bight. It would be interesting to discuss how the changes in chlorophyll-a may affect the food web structure, the carbon fluxes, and the ecological status of the region, and to compare the results with other studies in similar or contrasting regions.

- Line 494: Balkoni et al (in prep.)?!

- Although the work is very well written, a linguistic check would be very helpful, especially with the very long sentences.

---

## Referee Comment (RC2)

**Review of MS "Analyses of sea surface Chlorophyll-a trends and variability in a period of rapid Climate change, German Bight, North Sea ", from Felipe de Luca Lopes de Amorim et al.**

**General remarks**

- The study is relevant and important wrt marine ecosystems in the context of the climate trends, and the method is adequate to explore the questions.
- Yet, I have problems to identify the take-home message. The paper is very long and contains 18+ graphs, which somehow blurs the message.
- The Authors may want to provide clearer explanations about some statistical methods (e.g., combined EOFs and PCs results are not always straightforward to interpret). Clearly, these statistical approaches are rich and provide good insights, but they remain somewhat cryptic still. Such paper may be the opportunity to share knowledge and familiarize the community on the used methods. This is a non-mandatory suggestion.
- The language of the whole manuscript should be screened by an English-speaking colleague before publication (and, ideally, even before submission). I pinpointed some disturbing examples but did not underline all instances.

**Abstract**

This section describes with too many details the results, and could probably be shortened with a better summary of the results. What is the take-home message?

There are some unclear sentences that should be corrected. For instance:
- L27. "The monthly chlorophyll-a concentration anomalies covaried 45% with sea surface temperature anomalies" should better be "Monthly chlorophyll-a concentration anomalies covaried by 45% with sea surface temperature anomalies"
- L28-29. "This study demonstrated that the […] product can assess mostly of the known processes" should be "This study demonstrated that the […] product can evaluate most known processes"

**Introduction**

1. L57 'dimension' instead of 'domain'?
2. L59 'enabling the assessment of Chl-a spatiotemporal variability.' … but only at the surface.
3. L69-73 exhibit an argument that is between a discussion and an introduction. Having read it as it is written now, I am not sufficiently convinced that the approach is without flaws, as more questions are raised than answered. For instance, you mention a remote sensing (RS) sampling at depths comprised within 1-12 m (depending on turbidity). However, considering the total depth at the Helgoland sampling site (~6-10 m) and its surroundings sampled by satellite (~30-40 m in the Elbe Glacial Valley), we see that these are different depths. Is there a difference wrt the interpretation of the RS signal of Chl? I mean, if the satellite Chl is calibrated at

Helgoland sampling site, is it valid at deeper sites? And what do you do when the water column is stratified in summer (is it?)? When it is not stratified, it is well-mixed for dissolved substances, but not for particles (you even suggest this idea when you rightfully mention that turbulent mixing may enhance resuspension). What about that when it comes to analyze RS Chl signal? Do changes in turbulence only generate a small variability in Chl wrt the seasonal variability? Perhaps the Authors might want to be more affirmative in the Introduction (i.e., suggest less questions), and then discuss the details about RS signal, depth, stratification, resuspension, etc. in the Discussion? As far as I can see, it seems to be just a matter of presenting the argument.

4. L75 'Chlorophyll-a (Chl-a)' This acronym was already defined above. Please, double check the whole manuscript for overall consistency.

**Methods**

1. L136 '60 km of the German coast' Do you mean '60 km **off** the German coast'?
2. L138 'The samples are representative for the whole water column due to the well-mixed conditions'. Indeed, Wiltshire et al. say it in their paper of 2009 based on an earlier reference. Yet, isn't there a vertical gradient of particles (Chl and SPM) in spite of the vertical mixed conditions? Is it negligible for the purpose of this study?
3. L167 'As a pre-analysis, we calculated temporal mean and standard deviation (std) of the Chl-a anomalies.' When writing 'temporal mean' (or std) do you mean 'yearly mean' (or std)? Please, specify here.
4. If you see Fig.3b, would you consider that Chla anomalies are normally distributed, or skewed? Is it important when calculating the mean and std?
5. L175 '1 time step lagged' Is the lag one month, or is it another time length?

**Results**

1. L190 'Both time series showed significant negative trends, evaluated by the Mann Kendall trend test.' Difficult to see how this statement relates to Fig.2. It seems better linked to Fig.10…
2. Fig.3b Is the green colour the superimposition of both in situ and RS Chl? Please, clarify or improve the plot.
3. Fig.4d There is an increasing trend of Chla at the coast and a decreasing trend offshore. While any potential eutrophication/de-eutrophication trend may affect Chla, it would do it at the coast mainly. This is a very interesting result as it suggests that the (de-)eutrophication trend is not the only (or even the main) controlling factor of the Chl trend. This result motivates the study.
4. Fig.4 & 5 In this approach, attention is given to the spatial variability of Chl. It raises the question of whether the observed increasing trend in SST is also variable in space, or if it is homogenous in the G. Bight…
5. Fig.6 caption. The last sentence of the caption should be in the text, not in the caption.
6. L256 'bellow' => 'below' Please, check the MS for this kind of misprint.
7. L260-261 'although the spatial averaged Chl-a remote sensing was overestimated during winter months, and the second bloom peak was delayed in offshore areas.' Dubious interpretation. It seems the Authors were expecting the same results for

mean coastal RS Chl and Helgoland in situ Chl profiles. I do not see an 'overestimate' or a 'delay'. Profiles are just different.

8. Fig.8 caption. Once again, clarify please. Understanding what is on a graph should be made easy by the Authors for the reader, especially in a paper showing 18+ graphs. An effort should definitely be provided on that aspect.

9. Fig.8 Maybe I did not fully understand the EOF approach, but it is unclear to me why PC2 was averaged over the entire area and not over the two different areas (red and blue) identified with EOF2. As a side remark, PC2 shows a seasonal profile that reminds me the profile of SPM concentration in most coastal zones of the southern North Sea (high winter values, and low summer values due to TEP-enhanced flocculation of SPM).

10. L316 'The peak in Chl-a anomalies in 2008 was related with a positive peak of North Atlantic Oscillation index winter mean (NAO)' (and sentences next to it). This is not a convincing demonstration. I would be convinced if Chl anomalies in April were in general more correlated with winter NAOi. But I do not think it is the case. Therefore this statement seems very dubious to me. This being said, I have nothing against dividing the period into two segments around 2010, as the Authors did. These two periods seem indeed different wrt their mean April Chl, for instance. Some impartial statistical tests might even be conducted to justify this separation.

11. L329 'These results could be the response of earlier spring blooms in the period 2010-2020 compared to the years before.' Indeed, the results from March to May might indicate a forward shift of the spring bloom to earlier days in recent years. Did the Authors also have a look at the February distributions?

12. L347-361 Interesting results! Yet, I find it odd that the Authors offer an interpretation of why coastal Chl anomalies tend to increase in recent years without even mentioning a possible trend in coastal nutrients (or adjacent river loads, at least the Elbe)…

13. Fig.16 caption. Now, we know that the lag is one month… It should have been said in Methods (or perhaps I missed it?)

14. Fig 17 & 18. Improve caption please.

15. L492-3 is a direct repetition of L480-1

16. L494-5 The information about nutrients comes much too late in this manuscript about Chla variability. I wonder if it shouldn't even take place in the introduction as it is not a proper result of the study and nevertheless an important element of the story.

17. L499 'decreasing trends and slight increase of Chl-a' Unclear sentence.

18. The discussion does not discuss the validity of the approach. It is not always mandatory but in this case it may be more convincing (see, e.g, my comment in the Introduction section).

19. The conclusion seems a repetition of the Discussion with more numbers and less references. Where is the core message? When I see the results, I see a potential story. However, I do not find that story in the text.

---

## Author Response (AR1)

**Reply to Referee #1**

We thank the Anonymous Referee #1 for the effort in reviewing the manuscript and for her/his positive evaluation. The posted comments and suggestions helped us to improve the manuscript.

**Review of ''Analyses of sea surface Chlorophyll-a trends and variability in a period of rapid Climate change, German Bight, North Sea''.**

**In this work, the authors provide a comprehensive analysis of sea surface chlorophyll-a trends and variability in the German Bight, a coastal area in the North Sea, using satellite remote sensing data and in situ measurements. The paper aims to understand the relationship between chlorophyll-a, sea surface temperature, and mixed layer depth. The paper presents some interesting and novel findings, such as the significant positive trend of chlorophyll-a near the Elbe estuary and the negative trend in most of the central German Bight, the changes in the distribution of chlorophyll-a anomalies before and after 2009, and the contrasting modes of co-variability between chlorophyll-a and sea surface temperature or mixed layer depth in coastal and offshore areas. Overall, the paper is well-written and structured, but I think some of the figures could be improved before publication. I have some major/minor comments and suggestions, but I could not see any scientific flaws, and think the manuscript is a good addition to the field.**

Many thanks to the Reviewer for her/his time and effort to provide us with comments, they are valid and very helpful. Below, you will find our responses to each comment. The comments received concerning language are all accepted and changed accordingly in the main text; therefore, they are not further discussed.

1. **In this MS, the authors have used several statistical methods such as EOF, MCA, linear correlation, trend analysis, probability density function, and different types of tests such as the Mann-Kendall trend test, Kolmogorov-Smirnov test, and two-sided Wald test. However, the authors do not examine/illustrate these statistical techniques in detail. For example, what is the LOWESS trend, and how can it be estimated?**

The LOWESS is a non-parametric fitting technique, no assumptions about data distribution are necessary. It gives a better overview of trends in data with complex patterns. In the case of temperature, we see that we have periods with different linear trends, so the lowess method gives us this overall trend considering all these different periods. The trade-off is that lowess in more computational expensive, and doing this analysis for the whole gridpoints would be very time consuming. We included the original reference of Cleveland (1979) and Cheng et al. (2022) describes the usefulness of the technique.

Line 234: "Specifically within the German Bight, the mean SST anomaly trend, as estimated by the locally weighted scatterplot smoothing method (LOWESS; Cleveland, 1979; Cheng et al., 2022) indicated an increase of 0.77°C from 1998 to 2020 (Fig. 5)."

**What is the two-tailed Wald test? Is it different from the t-test? Could you please add more details about this test? What is the difference between this test and the Mann-Kendal trend test? Also, I strongly recommend using the modified Mann-Kendal test (Hamed and Ramachandra Rao, 1998), which takes into account the serial correlation between observations.**

**https://doi.org/10.1016/S0022-1694(97)00125-X**

The two-tailed Wald test is inherent from the tool used to calculate the linear trends, and following the                                                  description                                                  in https://docs.scipy.org/doc/scipy/reference/generated/scipy.stats.linregress.html. It is applied with t-distribution to compute the p-values.

The Mann Kendall test was used as a more robust calculation of trend significance, as it was already used in several works. The linear trends had the aim to give the first descriptive results about the spatial chl-a in the German Bight. Besides the Hamed and Rao modified Mann Kendall test, there is also the Yue and Wang modified Mann Kendall test (Yue and Wang, 2004), which agrees better in the coastal areas with the original Mann Kendall test. Considering that the Yue and Wang also corrects for serial autocorrelation, we estimated the trends significance using the Yue and Wang modified Mann Kendall test (Fig. 6 of the revised manuscript).

[Figure]

Yue, S., & Wang, C. (2004). The Mann-Kendall test modified by effective sample size to detect trend in serially correlated hydrological series. *Water resources management*, 18(3), 201-218. doi:10.1023/B:WARM.0000043140.61082.60

**The authors should also provide more details and justifications for some methodological choices, such as the definition of coastal and offshore regions, the criteria for significance tests and confidence levels, and the number of modes used for the EOF and MCA analyses.**

Thank you very much. We added the requested information in the Methodology Section.

Line 148: For analysis, coastal and offshore areas were defined by the isobaths of 30 m, following the description results obtained by the temporal mean and standard deviation, where areas with Chl-a mean higher than 1 mg m$^{-3}$ and standard deviation higher than 2 mg m$^{-3}$ define coastal areas (see Figure S1 in the Supplementary material). Consequently, the shallow Dogger Bank was considered in the offshore region.

The significance is based in the p-value obtained by the significance tests and considers the p-values lower than 0.05 as significative (95% confidence level). We added the following sentence in the text:

Line 312: The significances are based in p-values lower than 0.05 (95% confidence level).

The number of the EOF and MCA modes is based in the curve of explained variance, defining the inflexion of the curve as a limit for significant modes. We added the following sentence to the text:

Line 328: The inflexion of the explained variance curve defined the limit for significant EOFS modes.

**2. The MS is overloaded with content, analyses, and details, which can be reduced in some places for better understanding and easier to follow at times. For example, figure 4D and figure 12 are identical because the authors have already superimposed the significant and non-significant regions in Figure 4D.**

Thank you very much. The idea of figure 4D is to give the first description of the Chl-a trends in the German Bight, while figure 12 is a more robust assessment of the statistical significance of the observed trends using the Mann Kendall trend test.

**In addition, the entire section (Section 3.6) in the description of Figure 4D can be moved to the main body of Section 3.2 so as not to interrupt the story. In another example, from Figure 6 and Figure 7 and their description, the authors came to the same conclusion that the highest chlorophyll concentrations are found in the coastal region in April and May. And so on ...**

Thank you, we accepted the Reviewer's suggestion. Section 3.6 was merged with Section 3.1. We changed the 3.1 heading for "General findings and Chl-a overall trends".

**I strongly recommend adding a file of supplementary material that includes these figures (6 and 12) and others that are not discussed in detail in the main body of the MS (e.g., fig. 13).**

Thank you for the suggestion and we accept it as it will improve the clarity of our manuscript. Figures 7, 8 and 10 are in Supplementary material.

**3. The MS does not provide a clear explanation for the choice of 2009 as the breakpoint for the analysis of chlorophyll-a anomalies distribution. It seems that this year was selected based on the peak of chlorophyll-a anomalies observed in 2008, but the paper does not discuss the potential causes or implications of this peak.**

The reviewer is right that the year was selected based on the Chl-a peak observed in 2008. We discuss that this follows the positive NAO winter index pattern and positive peak, but we could not find a possible reason rather than a change in hydrography in the German Bight related to the increased inflow of Atlantic water. As it was a peak event that did not repeat during the analysed years, we prefer to not speculate without deeper analysis.

**It would be helpful to provide more justification and context for this choice and to explore the sensitivity of the results to different breakpoints. To detect the abrupt change in chlorophyll-a concentrations, I highly recommend using the Pettitt homogeneity test (Pettitt, 1979).**

**https://doi.org/10.2307/2346729**

Thank you, for the suggestion. We applied the Pettitt test to detect the change points in Chl-a time series for March, April and May in coastal and offshore regions. April and May Chl-a offshore, and also May Chl-a coast showed 2010 as the possible shift point. For March, both Chl-a coast

and offshore showed 2007 as probable change point. For April Chl-a coast, the year of 2012 was defined as probable shift point.

**4. Some parts of the paper are a repetition of the others, for example, Figure 8b does not bring any new results than those in Figure 7.**

The idea of Figure 8a and b is to test the EOF analysis for processes related to Chl-a variability that we already know, i.e. seasonal variability. In this case, first mode is explained by the two phytoplankton blooms observed in the region and the second mode is the decrease of phytoplankton during summer and winter.

**Also, I wonder why the authors estimated the seasonal cycle of each principal component at the seasonal (Figure 8b) and interannual scale (Figure 15 E, F, G, and H) although it is supposed to use the PCs to look at variability during the whole study period.**

To clarify, we applied the EOF analysis to seasonal Chl-a (monthly climatological means) and Chl- anomalies (seasonal signal removed by subtracting the climatological monthly means from the absolute Chl-a concentration). Figure 8 (bottom) is the temporal pattern of the second mode (PC2) applied to the Chl-a climatological means. Figure 8 is now part of the Supplementary material.

**In my opinion, Figure 8 does not provide any new results and can be part of the supplementary material.**

We moved Figure 8 to Supplementary material.

**In particular, the authors have already applied the EOF to the Chl-a anomalies (Figure 14 and Figure 15). Also, all spectral analyses applied to each principal component (Figure 15**

**I, J, K, and L) could be removed and applied the spectral analyses to the original data (Chl-a).**

We applied the spectral analysis to each of the PCs because they represent the temporal modes of variability for each of the spatial modes (EOFs). We could calculate the averaged spatial mean and apply the spectral analysis, but we believe information would be lost due to the lack of the spatial component.

**5.   The authors mention "a period of rapid climate change" in the MS title. It is not clear to me whether the authors consider the whole study period as a rapid climate change or whether they defined this period in their MS using a specific test. Please add more details on this point or support it with a reference in MS or change the title.**

The title is based on the findings of Amorim and Wiltshire et al. (2023) and also the results showed in Figure 5 of the manuscript. We followed the Reviewer's suggestion and changed the title to:

Line 1: Analyses of sea surface chlorophyll-a trends and variability from 1998 to 2020, German Bight, North Sea.

**6.   Objectives: In Lines 80-85 the five main objectives of the study are stated. For me, objectives (ii), (iii), and (iv) seem to be identical to the main objective (line 75). I would suggest rephrasing/rewriting the main goals concisely and clearly. I would also suggest that the authors put these in the final section of the paper when summarizing their findings in the conclusion. What is the difference between objectives (ii) and (iv)?**

Thank you. We rephrased the objectives and now they are as follow:

Line 197: (i)    The long-term trends of Chl-a.

(ii)     The dominant modes of Chl-a variability.

(iii)    The relationship between Chl-a, SST, and MLD in the region.

We also included in the Conclusions:

Line 814: "For objective i)…"

Line 822: ."In objective ii) …"

Line 829: "As objective iii) …"

**7. Lines 233-237: In this section, more details on the SST time series in Figure 5 are needed, e.g. which year has the highest and lowest SST anomalies and variability. In addition, the SST trend values obtained should be compared with previous studies in the same region to highlight differences and similarities.**

Thank you for this suggestion. We added the references pointed by the Reviewer and compared with the values observed in this study and Amorim and Wiltshire et al. (2023).

Line 402: The maximum and minimum SST anomalies were observed in July 2006 and April 2013, respectively. The maximum SST anomaly was 1.86 °C and the minimum was -2.8 °C. Interestingly, after the observed minimum in 2013, a sharp increase to positive SST anomalies was observed , initiating a period with strong positive trends.

**Furthermore, I suggest creating the spatial trend maps of SST. This will give the reader a clear picture of the spatial and temporal variability of SST trends in different locations of the study area, which can be compared to the chlorophyll trend map.**

**https://doi.org/10.3389/fmars.2023.1258117**

**https://doi.org/10.5194/nhess-22-1683-2022**

We created the SST anomalies trend map and estimated the significance using the modified Mann Kendall test. The results are homogeneous in the whole German Bight for the analysed period. We included the results in the Supplementary material.

**Other comments**

- **Figure 2: For the comparison, it would be better to draw a two-line time series in one panel instead of drawing the positive and negative anomaly for each one, which makes the comparison unclear.**

Thank you, we accepted the suggestion.

- **Figures 15,17, and 18, Shaded regions make these figures unclear, I suggest removing these shaded regions.**

We accepted the suggestion.

- **As far as I know, it would be better to limit the acronyms in the abstract and introduce them in the text (from the introductory chapter onwards).**

Thank you very much and we accepted the suggestion.

- **The abstract is very long and contains a more general and longer sentence, which can be shortened or moved to another section (e.g., introduction). For example, a sentence starts in line 12 and ends in line 15. The same for the next one (lines 15-18). Please try to shorten the abstract to be concise and focus on the most interesting results, of which there are many in your MS.**

Thank you very much. We present an improved and shorter Abstract in the revised PDF.

Line 10: Satellite remote sensing of ocean colour properties allows observation of the ocean with high temporal and spatial coverage, facilitating the better assessment of changes in marine primary production. Ocean productivity is often assessed using satellite derived chlorophyll-a concentrations, a commonly used proxy for phytoplankton concentration. We used the Copernicus GlobColour remote sensing chlorophyll-a surface concentration to investigate seasonal and non-seasonal variability, temporal trends, changes in spring bloom chlorophyll-a magnitude. Complementary, we analysed the chlorophyll-a relationship with sea surface temperature and mixed layer depth in the German Bight from 1998 to 2020. Empirical Orthogonal Functions were employed in order to investigate dominant spatial and temporal patterns (modes) related to the main processes of chlorophyll-a variability. Multi Covariance Analysis was used to extract the dominant structures that maximize the covariance between chlorophyll-a and sea surface temperature|mixed layer depth fields. High levels of Chl-a were found near the coast, showing a decreasing gradient towards offshore waters. A significant chlorophyll-a positive trend was observed close to the Elbe estuary and adjacent area, while 55% of the German Bight was characterized by a significant chlorophyll-a negative trend. The chlorophyll-a non-seasonal variability showed that the first four modes explained around 45% with the first and second modes related to inter and intra-annual variability, respectively, observed in the temporal principal components spectral analyses. Monthly chlorophyll-a concentration anomalies covaried by 45% with sea surface temperature anomalies and 23% with mixed layer depth anomalies. The monthly averages of chlorophyll-a anomaly fields were suitable to investigate long-term trends and variability. The rising water temperature, combined with its indirect effects on other variables, can partially explain the observed trends in chlorophyll-a.

- **Line 13: Please use "comparing with the in-situ data" instead of "comparing with the Helgoland Roads Chl-a in situ data".**

When shortening the Abstract, we removed this part of the sentence.

- **Line 19, "A significant long-term positive trend was observed close to the Elbe estuary and adjacent area". The trend of what?**

Thank you, we included "chlorophyll-a".

- **Please indicate the source of the bathymetry data used in Figure 1.**

Thank you. We included the GEBCO team reference.

Line 90: "Bathymetry of the German Bight (GEBCO Bathymetric Compilation Group 2023, 2023)."

- **Line 96: Please add the position of the Elbe estuary in Figure 1.**

Elbe position added to Figure 1.

- **Line 100: please use "flow" instead of "inserted"**

Change done.

- **Line 114: Please provide the doi and a reference to the data, if possible, instead of using the general link of CMEMS and the product name. Especially, the same link has been repeated in line 120 and line 125. Also, I suggest removing Table 1.**

Thank you, we included the doi and references and added the Data Availability Section.

Line 840: North Atlantic Chlorophyll (Copernicus-GlobColour) from Satellite Observations: Daily Interpolated (Reprocessed from 1997) replaced on July 2022 by the Atlantic Ocean Colour (Copernicus-GlobColour), Bio-Geo-Chemical, L4 (daily interpolated) from Satellite Observations

(1997-ongoing), E.U Copernicus Marine Service Information (CMEMS), Marine Data Store (MDS), DOI: https://doi.org/10.48670/moi-00289 (Accessed on 12 Dec 2021).

ESA SST CCI and C3S reprocessed sea surface temperature analyses. E.U. Copernicus Marine Service Information (CMEMS). Marine Data Store (MDS). DOI: https://doi.org/10.48670/moi-00169 (Accessed on 23-Mar-2023).

Atlantic- European North West Shelf- Ocean Physics Reanalysis. E.U. Copernicus Marine Service Information (CMEMS). Marine Data Store (MDS). DOI: https://doi.org/10.48670/moi-00059 (Accessed on 20-Apr-2023).

- **Line 120: which products are used for SST and MLD? It is not clear to me. Please add more details about these products.**

We clarified this in the Methodology.

Line 244: The SST dataset (ESA SST CCI and C3S reprocessed sea surface temperature analyses; https://doi.org/10.48670/moi-00289) contains gap-free maps of daily average SST at 0.05deg. horizontal grid resolution. It is a composition of satellite data from the Advanced Along-Track Scanning Radiometer (AATSR), Sea and Land Surface Temperature Radiometer (SLSTR) and the Advanced Very High Resolution Radiometer (AVHRR) (Lavergne et al., 2019; Merchant et al., 2019; Good et al., 2020).

Daily mixed-layer depth data (≈7 km horizontal resolution) are part of the Atlantic-European North West Shelf-Ocean Physics Reanalysis product (https://doi.org/10.48670/moi-00059). The MLD was defined as the depth where the increase in density, compared to the density at 3 m depth, corresponds to a temperature change of 0.8°C (Kara et al., 2000; PUM, 2021).

The NAO winter index data was obtained from the Climate Analysis Section, National Center for Atmospheric Research (NCAR, Hurrel et al., 2023).

- **Line 163 ''the two-sample Kolmogorov Smirnov test.'' please add the reference for this test.**

Reference added.

Massey, F. J. (1951). The Kolmogorov-Smirnov Test for Goodness of Fit. *Journal of the American Statistical Association*, *46*(253), 68–78. https://doi.org/10.1080/01621459.1951.10500769

- **Line 189: I suggest removing the acronym HPLC from title 3.1. I understand that it was used previously and refers to "high performance liquid chromatography" but should not be used in the title.**

Thank you. We followed the reviewer's suggestion.

**Line 189: Evaluation of in situ and Remote Sensing Chlorophyll-a**

Thank you, we removed it.

- **Line 190: "Both time series showed significant negative trends". Please add the values of these trends.**

We added the trend values of -0.031 and -0.025 mg m$^{-3}$ per year for in situ and remote sensing, respectively.

Line 336: "Both time series showed significant negative trends (in situ = -0.031 mg m$^{-3}$ y$^{-1}$ and remote sensing = -0.025 mg m$^{-3}$ y$^{-1}$), ..."

- **Lines 214-226: please refer to fig4b, fig4c, and fig4d in this section.**

Done.

- **Line 229: In the caption of Figure 4, I think the authors should use the spatial mean instead of the temporal mean. Or they can use spatial climatological means.**

Thank you for pointing this. We refer as temporal mean/std because is the mean/std in the time dimension of a spatial data. If it is confusing and the Reviewer prefers the term "climatological", we would change without hesitation.

- **Line 236: "However, when it comes to the averaged MLD, no significant trend was observed." On what basis do the authors come to this conclusion? Do they estimate the trend of temperature at MLD?**

No, we estimated the significance of the MLD trends as we did with SST (Fig. 5), but because it was not significant, we did not show. Besides, the spatial MLD trend analysis with Mann-Kendall test (not shown), did not give significant trends.

- **Line 290: I suggest starting the sentence with something else instead of the number.**

Thank you. We modified the text.

Line 290: "Considering the German Bight area here analysed, 96% had a maximum…"

- **Lines 402-408: What if the authors apply spectral analysis to the original data? Do they expect to get the same results?**

We applied the spectral analysis to each of the PCs because they represent the temporal modes of variability for each of the spatial modes (EOFs). We could calculate the averaged spatial mean and apply the spectral analysis, but we believe information would be lost due to the lack of the spatial component.

- **Figure 16; please use an appropriate range for the color bar, say between -0.4 and 0.4. It is not clear how the trends are significant in some regions and not significant in others, while both have the same trend values. Have you tried testing these correlations with different time lags and not just one month?**

Thank you, we changed the colour bar range. No, we did not apply longer time lags because, by our knowledge, scales longer than one month, in an intrannual scale, will decrease in correlation.

[Figure]

- **Please move lines 432-439 to the methodology section.**

Thank you, we accepted the suggestion.

- **The MS does not provide a clear link between the observed chlorophyll-a trends and variability and the broader implications for the marine ecosystem and biogeochemical cycles in the German Bight. It would be interesting to discuss how the changes in chlorophyll-a may affect the food web structure, the carbon fluxes, and the ecological status of the region, and to compare the results with other studies in similar or contrasting regions.**

Thank you. In an attempt to include how Chl-a changes impact in a more general and holistic way the ecology in the German Bight, we compared with other studies in other regions to show how complex is to point out general impacts. This goes beyond the scope of our study, but it is clearly a discussion that cannot be left outside when discussing changes in Chl-a, consequently in marine primary production.

Line 815: The German Bight, with its shallow bathymetry, does not behave as other marine regions, like the Arabic and Japan Sea, or like the oceanic gyres, where the changes in MLD are shown to be an important factor in Chl-a changes, mostly related with stratification and sea level anomaly (Prakash et al., 2012; Signorini et al., 2015; Park et al., 2020). A study in the Bohai Sea by Fu et al. (2016) identified an opposed result compared to our study, with positive trends in off shore areas and negative trends in coastal areas. Dunstan et al. (2018) described in their work how highly spatially heterogeneous are the covariance of Chl-a and SST, pointing the importance of regional studies and the complexity of the subject.

- **Line 494: Balkoni et al (in prep.)?!**

We will ask the Editor's help to provide a better citation format when we discuss manuscripts in preparation.

- **Although the work is very well written, a linguistic check would be very helpful, especially with the very long sentences.**

Thank you very much. We will put more care on it.

**Reply to Referee #2**

We thank the Anonymous Referee #2 for the effort in reviewing the manuscript and for her/his positive evaluation. The posted comments and suggestions helped us to improve the manuscript.

**Review of MS "Analyses of sea surface Chlorophyll-a trends and variability in a period of rapid Climate change, German Bight, North Sea ", from Felipe de Luca Lopes de Amorim et al.**

Many thanks to the Reviewer for her/his time and effort to provide us with comments, they are valid and very helpful. Below, you will find our responses to each comment. The comments received concerning language are all accepted and changed accordingly in the main text; therefore, they are not further discussed.

**General remarks**

**• The study is relevant and important wrt marine ecosystems in the context of the climate trends, and the method is adequate to explore the questions.**

Thank you very much.

**• Yet, I have problems to identify the take-home message. The paper is very long and contains 18+ graphs, which somehow blurs the message.**

Thank you, we tried to reduce the amount of information focusing in the most important results and moved some of the graphs to Supplementary material.

Our take-home message, considering the dominant decreasing trends in chlorophyll-a concentration observed in this study, is that in a shallow German Bight, the impact of increasing temperatures are more important than stratification in the chlorophyll-a variability. However, these temperature effects are mostly indirect, and temperature is an indicator of hydrographic changes in the German Bight. We tried to improve the clarity of our take-home message and included this answer to the Conclusions.

**• The Authors may want to provide clearer explanations about some statistical methods (e.g., combined EOFs and PCs results are not always straightforward to interpret). Clearly, these statistical approaches are rich and provide good insights, but they remain somewhat cryptic still. Such paper may be the opportunity to share knowledge and familiarize the community on the used methods. This is a non- mandatory suggestion.**

We have a paragraph in the text (Lines 374-380) explaining a general analysis of EOF results. Because the EOF is completely mathematical, the researchers do the physical interpretation, and it is related to their pre-knowledge about the variability processes they want to study.

**• The language of the whole manuscript should be screened by an English-speaking colleague before publication (and, ideally, even before submission). I pinpointed some disturbing examples but did not underline all instances.**

We are sorry and increased our effort in the text.

**Abstract**

**This section describes with too many details the results, and could probably be shortened with a better summary of the results. What is the take-home message? There are some unclear sentences that should be corrected. For instance:**

**• L27. "The monthly chlorophyll-a concentration anomalies covaried 45% with sea surface temperature anomalies" should better be "Monthly chlorophyll-a concentration anomalies covaried by 45% with sea surface temperature anomalies"**

Thank you, we accepted the suggestion.

**• L28-29. "This study demonstrated that the […] product can assess mostly of the known processes" should be "This study demonstrated that the […] product can evaluate most known processes"**

Thank you, we accepted the suggestion.

**Introduction**

**1. L57 'dimension' instead of 'domain'?**

Thank you, we accepted the suggestion.

**2. L59 'enabling the assessment of Chl-a spatiotemporal variability.' … but only at the surface.**

Yes, only in the surface. We added "surface" to the following sentence.

Line 59: Satellite data offers a solution to this problem by providing comprehensive spatial and temporal coverage, enabling the assessment of surface Chl-a spatiotemporal variability.

**3. L69-73 exhibit an argument that is between a discussion and an introduction. Having read it as it is written now, I am not sufficiently convinced that the approach is without flaws, as more questions are raised than answered. For instance, you mention a remote sensing (RS) sampling at depths comprised within 1-12 m (depending on turbidity). However, considering the total depth at the Helgoland sampling site (~6-10 m) and its surroundings sampled by satellite (~30-40 m in the Elbe Glacial Valley), we see that these are different depths. Is there a difference wrt the interpretation of the RS signal of Chl? I mean, if the satellite Chl is calibrated at Helgoland sampling site, is it valid at deeper sites? And what do you do when the water column is stratified in summer (is it?)? When it is not stratified, it is well-mixed for dissolved substances, but not for particles (you even suggest this idea when you rightfully mention that turbulent mixing may enhance resuspension). What about that when it comes to analyze RS Chl signal? Do changes in turbulence only generate a small variability in Chl wrt the seasonal variability? Perhaps the Authors might want to be more affirmative in the Introduction (i.e., suggest less questions), and then discuss the details about RS signal, depth, stratification, resuspension, etc. in the Discussion? As far as I can see, it seems to be just a matter of presenting the argument.**

Thank you for this suggestion. It would have been interesting to explore this aspect. However, in the case of our study, it seems slightly out of scope because we assume that the monthly means will remove the stratification effect, so we can consider a Chl-a response as the one observed in a vertically homogeneous water column. We hope this simple argument will be enough to answer the Reviewer's comment.

**4. L75 'Chlorophyll-a (Chl-a)' This acronym was already defined above. Please, double check the whole manuscript for overall consistency.**

Thank you for pointing this. We checked it.

**Methods**

**1. L136 '60 km of the German coast' Do you mean '60 km off the German coast'?**

Yes, thank you. Fixed.

**2. L138 'The samples are representative for the whole water column due to the well-mixed conditions'. Indeed, Wiltshire et al. say it in their paper of 2009 based on an earlier reference. Yet, isn't there a vertical gradient of particles (Chl and SPM) in spite of the vertical mixed conditions? Is it negligible for the purpose of this study?**

Thank you very much for this question. We used the monthly means of Chl-a surface exactly to overcome this problem with vertical gradients. Considering the time scale of mixing processes in the German Bight, the monthly means make the vertical gradients negligible.

**3. L167 'As a pre-analysis, we calculated temporal mean and standard deviation (std) of the Chl-a anomalies.' When writing 'temporal mean' (or std) do you mean 'yearly mean' (or std)? Please, specify here.**

To clarify, we mean the temporal average of the whole analysed period, losing the temporal dimension and only keeping the spatial one.

Line 167: "As a pre-analysis, we calculated temporal mean and standard deviation (std) of the Chl-a anomalies for the whole analysed period."

**4. If you see Fig.3b, would you consider that Chla anomalies are normally distributed, or skewed? Is it important when calculating the mean and std?**

Considering that the frequency of the highest Chl-a anomalies is very little compared to the whole sampling, the highest density of values are still around the 0 mean. We could consider the Chl-a anomalies, if not normally distributed, weakly skewed, which the mean and standard deviation are still valid.

**5. L175 '1 time step lagged' Is the lag one month, or is it another time length?**

Yes, the lag is one month, clarified in Line 315 and Line 175.

Line 315: We examined the relationship between the Chl-a anomaly fields with SST and MLD anomalies applying linear correlations in the direct anomaly fields and in 1 time step (one month) lagged Chl-a in relation to SST and MLD.

Line 175: "…the direct anomaly fields and in lagged time step of one month Chl-a in relation to SST and MLD."

**Results**

**1. L190 'Both time series showed significant negative trends, evaluated by the Mann Kendall trend test.' Difficult to see how this statement relates to Fig.2. It seems better linked to Fig.10…**

We verified the values of the trends for both time series presented in Figure 2 using the modified Mann Kendall trend test, and both showed decreasing trends. Following comment from Reviewer #1, we changed Figure 2 and included the trend values for better visualization.

**2. Fig.3b Is the green colour the superimposition of both in situ and RS Chl? Please, clarify or improve the plot.**

Yes, it is the superimposition. We made clear in the caption.

Line 211: Figure 1: a) Boxplots and b) distributions of remote sensing (orange) and in-situ (blue) monthly Chl-a anomalies. The shaded areas in b) are the superimposition of in situ and remote sensing bars.

**3. Fig.4d There is an increasing trend of Chla at the coast and a decreasing trend offshore. While any potential eutrophication/de-eutrophication trend may affect Chla, it would do it at the coast mainly. This is a very interesting result as it suggests that the (de-) eutrophication trend is not the only (or even the main) controlling factor of the Chl trend. This result motivates the study.**

Thank you, we will consider this comment to improve the take-home message of the manuscript. Please, also refer to the answer on comment 12 of the Results.

**4. Fig.4 & 5 In this approach, attention is given to the spatial variability of Chl. It raises the question of whether the observed increasing trend in SST is also variable in space, or if it is homogenous in the G. Bight…**

It is homogeneous in the whole German Bight. We included an image in the Supplementary material.

**5. Fig.6 caption. The last sentence of the caption should be in the text, not in the caption.**

We accepted the request.

Line 245: It is possible to observe the intra-annual behaviour of Chl-a, with a positive gradient from open waters to coast, and the increase in Chl-a in April and August.

**6. L256 'bellow' => 'below' Please, check the MS for this kind of misprint.**

Thank you, we fixed it.

**7. L260-261 'although the spatial averaged Chl-a remote sensing was overestimated during winter months, and the second bloom peak was delayed in offshore areas.' Dubious interpretation. It seems the Authors were expecting the same results for mean coastal RS Chl and Helgoland in situ Chl profiles. I do not see an 'overestimate' or a 'delay'. Profiles are just different.**

We consider the in situ data to be characteristic of a transitional zone in the German Bight. When we mentioned overestimation and delay, we referred to the capacity of the in situ data from one location to represent the whole German Bight area. To avoid doubts, we removed part of the sentence.

Line 258: "The in situ HRTS acquired in the transitional zone of the German Bight, between coastal and offshore areas, aligned well with the spatial averages of Chl-a remote sensing."

**8. Fig.8 caption. Once again, clarify please. Understanding what is on a graph should be made easy by the Authors for the reader, especially in a paper showing 18+ graphs. An effort should definitely be provided on that aspect.**

Thank you for the valuable comment. We gave more attention to the captions. Figure 8 was moved to Supplementary material.

Line 285: "First and second EOF spatial pattern (top) and PC temporal modes (bottom) of monthly climatological means of Chl-a. Dashed thin line superimposed by the PC lines is the Chl-a spatially

averaged at the study area. March-April period (green shaded) and August-September (pink shaded)."

**9. Fig.8 Maybe I did not fully understand the EOF approach, but it is unclear to me why PC2 was averaged over the entire area and not over the two different areas (red and blue) identified with EOF2. As a side remark, PC2 shows a seasonal profile that reminds me the profile of SPM concentration in most coastal zones of the southern North Sea (high winter values, and low summer values due to TEP-enhanced flocculation of SPM).**

In Figure 8 (bottom), we show PC1 and PC2 temporal patterns, superimposed by the Chl-a concentration averaged for the whole area. This was made to show that what dominates the variability of seasonal Chl-a is the presence of two phytoplankton blooms (peaks) and the decrease of Chl-a during summer and winter months. This is exactly what the spatial average of Chl-a shows. The PC2 result shows that the decrease during summer and winter months is not the same for coastal and offshore areas, because the decrease during summer in coastal areas is not as significant as in offshore areas when compared with the spring phytoplankton bloom. As these results are well described in the literature, it validates the EOF variability analysis.

**10. L316 'The peak in Chl-a anomalies in 2008 was related with a positive peak of North Atlantic Oscillation index winter mean (NAO)' (and sentences next to it). This is not a convincing demonstration. I would be convinced if Chl anomalies in April were in general more correlated with winter NAOi. But I do not think it is the case. Therefore this statement seems very dubious to me. This being said, I have nothing against dividing the period into two segments around 2010, as the Authors did. These two periods seem indeed different wrt their mean April Chl, for instance. Some impartial statistical tests might even be conducted to justify this separation.**

Thank you very much. We conducted a Pettitt shift test suggested by Reviewer #1.

**11. L329 'These results could be the response of earlier spring blooms in the period 2010-2020 compared to the years before.' Indeed, the results from March to May might indicate a forward shift of the spring bloom to earlier days in recent years. Did the Authors also have a look at the February distributions?**

No, we did not look at the February distributions. We considered that light availability did not change in the analysed period and February primary production would still be limited by light.

Considering the Reviewer's #2 interest, we calculated the February Chl-a anomalies distributions:

[Figure]

What is observed is that in offshore areas, the distribution shifted to negative anomalies in the period of 2010-2020, characterized by values below the mean of the period 1998-2020. In the coast, the distribution in the period 2010-2020 increased variability for both negative and positive values (stretched in the x-axis), with a very week bimodal distribution due to a peak in positive anomalies.

**12. L347-361 Interesting results! Yet, I find it odd that the Authors offer an interpretation of why coastal Chl anomalies tend to increase in recent years without even mentioning a possible trend in coastal nutrients (or adjacent river loads, at least the Elbe)…**

This is a very interesting hypothesis proposed by the Reviewer #2. Schulz, G. (2023) shows that the amount of nitrogen being carried to the Elbe estuary (consequently the German Bight coast) decreased by both control in river eutrophication and less river discharge towards the estuary. The decrease in river discharge would also affect other essential nutrients like phosphate. Considering these findings, we could not consider that there is an increase in nutrients in the German Bight, at least not form the Elbe. I hope we understood the question and answered it properly.

Schulz, G., van Beusekom, J. E., Jacob, J., Bold, S., Schöl, A., Ankele, M., ... & Dähnke, K. (2023). Low discharge intensifies nitrogen retention in rivers–a case study in the Elbe River. Science of The Total Environment, 904, 166740.

**13. Fig.16 caption. Now, we know that the lag is one month… It should have been said in Methods (or perhaps I missed it?)**

Yes, the lag is one month. We clarified this in the Methodology.

**14. Fig 17 & 18. Improve caption please.**

We improved captions as shown below:

Line 441: "Figure 17: Results of MCA showing the first co-varying mode between Chl-a and SST anomalies. Top images are the co-variability mode maps (Chl-a: left; SST: right). Bottom images are the temporal co-variability PC1 (left) and the corresponding spectra (right). PC1 is shown as rolling means of 6 points to better visualize the temporal co-variability of mode 1 between Chl-a and SST anomalies."

Line 456: "Figure 18: Results of MCA showing the first co-varying mode between Chl-a and MLD anomalies. Top images are the co-variability mode maps (Chl-a: left; MLD: right). Bottom images are the temporal co-variability PC1 (left) and the corresponding spectra (right). PC1 is shown as rolling means of 6 points to better visualize the temporal co-variability of mode 1 between Chl-a and MLD anomalies."

**15. L492-3 is a direct repetition of L480-1**

Thank you very much for pointing this mistake. We removed the sentence from lines 480-481.

**16. L494-5 The information about nutrients comes much too late in this manuscript about Chla variability. I wonder if it shouldn't even take place in the introduction as it is not a proper result of the study and nevertheless an important element of the story.**

We accepted the suggestion and moved the sentence to the introduction.

Line 54: "Changes in nutrient concentrations have profound impacts on phytoplankton productivity and species composition (Hickel et al., 1993; Topcu et al., 2011; Burson et al, 2016). Balkoni et al (in prep.) estimated nutrients decadal changes in the German Bight and the results point out that there is a decrease in nutrients."

**17. L499 'decreasing trends and slight increase of Chl-a' Unclear sentence.**

Thank you for pointing this. We rephrased as:

Line 499: "Alvera-Azcárate et al. (2021) showed the heterogeneity of the North Sea, with areas presenting decreasing trends and others indicating slightly increasing trends of Chl-a."

**18. The discussion does not discuss the validity of the approach. It is not always mandatory but in this case it may be more convincing (see, e.g, my comment in the Introduction section).**

Thank you. Along the text, we cited different studies that made use of remote sensing, EOF and trend analysis, corroborating the validity of the approach used in our study.

**19. The conclusion seems a repetition of the Discussion with more numbers and less references. Where is the core message? When I see the results, I see a potential story. However, I do not find that story in the text.**

Thank you very much. We will try to improve the clarity of our take-home message as discussed in the General Remarks.